# Multi-Environment POMDPs: Discrete Model Uncertainty Under Partial Observability

**Eline M. Bovy**[*]
Radboud University
Nijmegen, The Netherlands
`eline.bovy@ru.nl`

**Caleb Probine**[*]
The University of Texas at Austin
Austin, TX, USA
`cprobine@utexas.edu`

**Marnix Suilen**
University of Antwerp – Flanders Make
Antwerp, Belgium
`marnix.suilen@uantwerpen.be`

**Ufuk Topcu**
The University of Texas at Austin
Austin, TX, USA
`utopcu@utexas.edu`

**Nils Jansen**
Ruhr-University Bochum & Radboud University
Bochum, Germany & Nijmegen, The Netherlands
`n.jansen@rub.de`

## Abstract

Multi-environment POMDPs (ME-POMDPs) extend standard POMDPs with discrete model uncertainty. ME-POMDPs represent a finite set of POMDPs that share the same state, action, and observation spaces, but may arbitrarily vary in their transition, observation, and reward models. Such models arise, for instance, when multiple domain experts disagree on how to model a problem. The goal is to find a single policy that is robust against any choice of POMDP within the set, *i.e.*, a policy that maximizes the worst-case reward across all POMDPs. We generalize and expand on existing work in the following way. First, we show that ME-POMDPs can be generalized to POMDPs *with sets of initial beliefs*, which we call *adversarial-belief POMDPs* (AB-POMDPs). Second, we show that any arbitrary ME-POMDP can be reduced to a ME-POMDP that only varies in its transition and reward functions or only in its observation and reward functions, while preserving (optimal) policies. We then devise exact and approximate (point-based) algorithms to compute robust policies for AB-POMDPs, and thus ME-POMDPs. We demonstrate that we can compute policies for standard POMDP benchmarks extended to the multi-environment setting.

## 1   Introduction

Partially observable Markov decision processes (POMDPs) [25] are important models for sequential decision-making under uncertainty. With numerous real-world applications, from robotics [47] to healthcare [21, 49], many algorithms to compute optimal policies have been proposed [32, 39, 40].

Planning algorithms that compute policies for POMDPs rely on knowing the exact parameters of the underlying transition and observation dynamics, an assumption that is often prohibitive in practice. Consider, for instance, a setting where a POMDP model is constructed by domain experts. A common

---

[*]Shared first authorship, ordered alphabetically.

39th Conference on Neural Information Processing Systems (NeurIPS 2025).

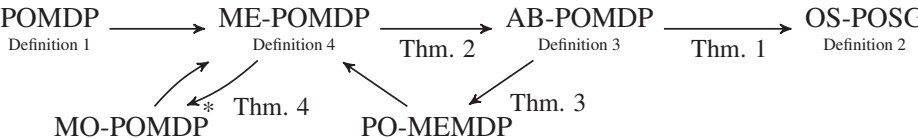

Figure 1: ME-POMDPs are a rich class of models between POMDPs and one-sided POSGs. Arrows from class A to class B indicate that we can transform models in class A to models in class B. The transformed model size is polynomial in the original model size for all arrows not marked by $*$. Unmarked arrows are trivial reductions. We define MO-POMDPs and PO-MEMDPs in Section 4.3.

example is the preservation of endangered bird species, akin to Chades et al. [8]. Multiple experts may disagree on parts of the model, leading to *discrete sets* of different transition and observation functions, and thus, a discrete set of POMDPs which we call a *multi-environment POMDP* (ME-POMDP). Without expressing any preference for one expert over another, *i.e.*, assuming a prior over the models in the ME-POMDP, a policy needs to be *robust* against all possible dynamics. That is, the policy needs to be optimized against the worst-case POMDP. We study ME-POMDPs and develop exact and approximate methods to compute optimal policies that maximize worst-case reward.

While multi-environment models have been studied extensively in the fully observable case, under the name of multi-environment MDPs (MEMDPs) [9, 35], existing algorithms do not apply to the partially observable case. MEMDPs are the discrete version of a broader class of models with continuous uncertainty known as *robust MDPs* [2, 24, 50]. *Robust POMDPs* [RPOMDPs; 5, 31] are the generalization of robust MDPs to the partially observable setting. While ME-POMDPs are, in theory, contained in RPOMDPs, algorithms for RPOMDPs rely on structural and semantic assumptions such as convexity, rectangularity, and dynamic uncertainty [5, 24, 50] as we discuss later. These assumptions make their application to ME-POMDPs unsuitable or overly conservative.

**Contributions** We study ME-POMDPs and devise algorithms to compute robust policies against any adversarial choice of POMDP in the ME-POMDP. We avoid overt conservativeness that appears when existing RPOMDP approaches are applied to ME-POMDPs. To summarize:

1. **Multi-Environment and Adversarial-Belief POMDPs** We generalize and expand on the theory of *multi-environment POMDPs*. We introduce *adversarial-belief POMDPs* (AB-POMDPs): POMDPs where the initial belief is adversarially chosen from a set of possible initial beliefs. We prove that AB-POMDPs are a special case of one-sided partially observable stochastic games (POSGs) [22], and show how any multi-environment POMDP can be modeled as an adversarial-belief POMDP. We also show that we can reduce any ME-POMDP to a restricted version where either the models do not differ in transitions, *i.e.*, a multi-observation POMDP (MO-POMDP), or do not differ in observation functions, *i.e.*, a partially-observable MEMDP (PO-MEMDP). We outline these relationships in Figure 1.
2. **Exact and approximate algorithms for AB-POMDPs** We prove that we can combine value iteration methods for POMDPs with linear programming to solve AB-POMDPs and thus ME-POMDPs. Specifically, we augment heuristic search value iteration (HSVI) with linear programming to get *AB-HSVI*, a point-based method for approximating value functions in adversarial-belief POMDPs. We evaluate AB-HSVI on standard benchmarks extended to the multi-environment setting and discuss how solving ME-POMDPs trades expected reward for computation time compared to a naive, non-robust baseline.

## 2 Related Work

Discrete model uncertainty has primarily been studied in the fully observable setting of multi-environment MDPs [MEMDPs; 35]. A large body of work assumes a distribution on the MEMDP's environments and provides algorithms for finding optimal policies [7–9, 12, 37, 43]. Recent work also studies reinforcement learning in MEMDPs where the environment distribution is unknown [11, 12, 27, 28, 51]. In contrast, we do not assume such a distribution exists, and instead focus on computing policies that are *robust* against all possible environments. For MEMDPs, the robust setting has been considered for *qualitative* objectives [10, 35, 46, 48], and integer programming has been used to find policies for robust reward maximization [1, 42]. *Robust MDPs* [24, 50] also capture the robust setting

but tend to assume continuous, compact, and convex model uncertainty. Moreover, most robust MDP algorithms require uncertainty to be independent across states or state-action pairs, an assumption known as *rectangularity*. The discrete uncertainty used in MEMDPs and our setting is inherently *non-rectangular*. See Suilen et al. [45] for a detailed discussion of rectangularity assumptions.

We combine discrete model uncertainty with partial observability. Robust POMDPs [RPOMDPs; 5, 31] extend robust MDPs with partial observability, but again rely on the assumption of convex and rectangular uncertainty sets [13, 16, 30, 31, 44]. Two RPOMDP papers address discrete settings [23, 36], however, these works assume the underlying model can change at each timestep, *i.e.*, dynamic uncertainty. In contrast, when we consider the worst-case model, we constrain the model to be consistent across timesteps. The work [15] develops subgradient descent algorithms for robust reward maximization in hidden-model POMDPs, a type of ME-POMDP.[2] In contrast, our algorithms are value-iteration-based, and we provide thorough characterizations of the relationships between different subclasses of multi-environment POMDPs, and their relationships to POSGs.

Work on unsupervised environment design, *e.g.*, [14, 29], uses underspecified POMDPs, equivalent to ME-POMDPs, as a model when designing reinforcement learning agents. In contrast to these works, which provide gradient-based learning approaches for minimizing regret, we give value-iteration-based planning methods for designing robust policies.

## 3 Preliminaries

A probability distribution on a finite set $X$ is a mapping $\mu\colon X \to [0,1]$ such that $\sum_{x \in X} \mu(x) = 1$. We denote the set of distributions on $X$ by $\Delta(X)$. Given two distributions $\mu_X$ and $\mu_Y$ on sets $X$ and $Y$, respectively, we denote their product distribution on $X \times Y$ by $\mu_X \times \mu_Y$. For $x \in X$, $\delta_x$ is the Dirac distribution that satisfies $\delta_x(x) = 1$. We denote the set of integers $\{1, \ldots, n\}$ by $[n]$. The set of finite sequences with elements in a set $C$ is $C^*$. We use $\perp$ and $\top$ for dummy states and observations.

We now introduce partially observable Markov decision processes (POMDPs). We consider expected cumulative reward maximization over both finite and discounted infinite horizons. For brevity, we compress these two settings into a single definition.

**Definition 1 (POMDP)** *A partially observable Markov decision process (POMDP) is a tuple $\mathcal{M} = (S, A, Z, T, O, R, b, \gamma, H)$ where $S$, $A$, and $Z$ are finite sets of states, actions, and observations, $T\colon S \times A \to \Delta(S)$ is a transition function, $O\colon S \times A \to \Delta(Z)$ is an observation function, $R\colon S \times A \to \mathbb{R}$ is a reward function, $b \in \Delta(S)$ is an initial state distribution, $\gamma \in [0,1]$ is a discount factor, and $H \in \mathbb{N} \cup \{\infty\}$ is a horizon.*

When $H = \infty$, we restrict $\gamma \in [0,1)$. For the case where $\gamma = 1$, we require that the horizon $H$ be finite. Additionally, for the case where $H = \infty$, we define $H + 1 = H$, such that $H + 1 = \infty$. Unless we specify otherwise, we assume all actions are available in all states.

A fully observable Markov decision process is a POMDP where $Z = S$ and $O$ is the deterministic identity mapping, *i.e.*, $\forall s \in S, a \in A\colon O(s,a) = \delta_s$. A policy $\pi$ in a POMDP maps a history of actions and observations to a distribution over the actions, *i.e.*, $\pi\colon (A \times Z)^* \to \Delta(A)$. We denote the set of policies in a POMDP $\mathcal{M}$ by $\Pi_{\mathcal{M}}$. We remark that one may differentiate between two classes of history-dependent randomized policies. A *behavioral policy* is a mapping $\pi\colon (A \times Z)^* \to \Delta(A)$, while a *mixed policy* is a distribution over deterministic behavioral policies.

A belief $b \in \Delta(S)$ describes the probability of being in a state given the initial state distribution and a history. We can define belief-based behavioral and mixed policies $\pi\colon \Delta(S) \to \Delta(A)$ and $\pi\colon \Delta(\Delta(S) \to A)$. Unless otherwise mentioned, we use history-based behavioral policies.

The value of policy $\pi$ in a POMDP $\mathcal{M}$ is $V_{\mathcal{M}}^{\pi} = \mathbb{E}\left[\sum_{t=1}^{H} \gamma^{t-1} r_t\right]$, where $r_t$ is the reward at time $t$. A policy $\pi^*$ is optimal if for all $\pi$, we have $V_{\mathcal{M}}^{\pi^*} \geq V_{\mathcal{M}}^{\pi}$. We denote the optimal value as $V_{\mathcal{M}}^* = V_{\mathcal{M}}^{\pi^*}$.

### 3.1 Solving POMDPs

Standard POMDP methods use piecewise-linear convex (PWLC) representations of value functions through a set of linear functions, known as $\alpha$-vectors [25]. Each $\alpha$-vector $\alpha\colon S \to \mathbb{R}$ represents a

---

[2]We remark that [15] was contemporaneous and leave an empirical evaluation against it as future work.

deterministic policy and maps states to the values of following that policy when we initialize the POMDP at that state. In the finite-horizon setting, $\alpha$-vectors represent $t$-step history-based policies, and we compute a new set of $\alpha$-vectors $\Gamma_t$ for each timestep $t \leq H$.

$$\Gamma_1 = \{\alpha \colon S \to \mathbb{R} \mid \alpha(s) = R(s,a), \forall a \in A\}, \tag{1}$$

$$\Gamma_t = \big\{\alpha \colon S \to \mathbb{R} \mid \alpha(s) = R(s,a) + \sum\nolimits_{(s',z) \in S \times Z} T(s,a)(s')O(s',a)(z)\alpha_z(s'), \tag{2}$$

$$\forall (a, \alpha_{z_1}, \ldots, \alpha_{z_{|Z|}}) \in A \times (\Gamma_{t-1})^Z\big\}.$$

The upper envelope of the $\alpha$-vectors in $\Gamma_t$ forms a PWLC function that corresponds to the optimal value function $V^*$ for horizon $t$. The optimal value given initial state distribution $b_0$ is given by $V^*(b_0) = \max_{\alpha \in \Gamma_t} \sum_{s \in S} b_0(s)\alpha(s)$.

In the infinite-horizon discounted setting, $\alpha$-vectors represent belief-based policies. Instead of computing multiple sets, we iteratively expand a single set $\Gamma$. The upper envelope of $\Gamma$ can approximate the optimal value function arbitrarily closely. For some initial set of $\alpha$-vectors $\Gamma$, we can compute a new $\alpha$-vector for each $(a, \alpha_{z_1}, \ldots, \alpha_{z_{|Z|}}) \in A \times \Gamma^Z$ similar to the finite-horizon setting as follows.

$$\alpha(s) = R(s,a) + \gamma \sum\nolimits_{(s',z) \in S \times Z} T(s,a)(s')O(s',a)(z)\alpha_z(s').$$

In both settings, we can prune $\alpha$-vectors from $\Gamma$ that are pointwise dominated by a single other $\alpha$-vector [38]. Pruning can be performed at any iteration, since pointwise dominated $\alpha$-vectors will never contribute to the upper envelope of $\Gamma$ or future iterations of $\Gamma$.

**Heuristic search value iteration** Various approximate POMDP solvers are based on $\alpha$-vectors, defining different ways to expand $\Gamma$ to efficiently approximate the optimal value function, such as [32, 41]. In particular, *heuristic search value iteration* [HSVI; 41] keeps track of both an upper and lower bound, *i.e.*, a set of belief value tuples and a set of $\alpha$-vectors, respectively, of the optimal value function. HSVI performs a depth-first search from an initial state distribution, updating the bounds along the way. The depth-first search selects actions optimistically with respect to the upper bound, and observations leading to the belief with the largest uncertainty, *i.e.*, the largest *gap* between the upper and lower bound. The depth-first search continues until the belief at depth $t$ has a gap of at most $\epsilon \cdot \gamma^{-t}$, with $\epsilon > 0$ a predefined error. After each depth-first search, the gap between the upper and lower bounds at the initial state distribution is computed. If the gap exceeds $\epsilon$, we continue with another depth-first search. The upper and lower bounds are initialized before the first depth-first search. HSVI computes the initial upper bound with the Fast Informed Bound [FIB; 20], and the initial lower bound with an $\alpha$-vector for each policy that always plays the same action [19].

### 3.2 One-sided Partially Observable Stochastic Games

Next, we introduce the specific form of partially observable stochastic games (POSGs) we consider in this paper. As with POMDPs, we again consider both finite horizon and discounted infinite horizon settings, and the same restrictions on $\gamma$ and $H$ apply.

**Definition 2 (POSG)** *A one-sided partially observable stochastic game (POSG) is a tuple $\mathcal{G} = (S, A_1, A_2, Z, T, O, R, b, \gamma, H)$ where $S$ is a finite set of states, $A_1$ is a finite action set for the partially observing player, $A_2$ is a finite action set for the fully observing player, $Z$ is a finite set of observations, $T \colon S \times A_1 \times A_2 \to \Delta(S)$ is the transition function, $O \colon S \times A_1 \times A_2 \to \Delta(Z)$ is the observation function, $R \colon S \times A_1 \times A_2 \to \mathbb{R}$ is the reward function, $b \in \Delta(S)$ is an initial state distribution, and $\gamma \in [0,1]$ and $H \in \mathbb{N} \cup \{\infty\}$ are the discount factor and horizon respectively.*

We consider concurrent POSGs, as studied in Horák et al. [22]. A policy for the partially observing player is a mapping $\pi_1 \colon (A_1 \times Z)^* \to \Delta(A_1)$, while a policy for the fully observing player is a mapping $\pi_2 \colon (S \times A_1 \times A_2 \times Z)^* \times S \to \Delta(A_2)$. We write $\Pi_{\mathcal{G}}^1$ and $\Pi_{\mathcal{G}}^2$ to denote the sets of policies for the partially observing and fully observing players, respectively.

A pair of policies $(\pi_1, \pi_2)$ defines a distribution on state-action trajectories in a POSG. We define the value of a policy $\pi_1$ for the partially observing player by the worst-case expected reward $V_{\mathcal{G}}^{\pi_1} = \min_{\pi_2 \in \Pi_{\mathcal{G}}^2} \mathbb{E}\big[\sum_{t=1}^H \gamma^{t-1} r_t\big]$, where $r_t$ is again the reward at time $t$. The value of the game is $V_{\mathcal{G}}^* = \max_{\pi_1} V_{\mathcal{G}}^{\pi_1}$. Existing algorithms for one-sided POSGs work by adapting exact and point-based value iteration techniques for POMDPs [22].

# 4 Adversarial-Belief and Multi-Environment POMDPs

We now formally introduce adversarial-belief POMDPs (AB-POMDPs) and multi-environment POMDPs (ME-POMDPs), and show the relations between those models and POSGs.

## 4.1 Adversarial-Belief POMDPs

*Adversarial-belief POMDPs* are POMDPs where we replace the initial belief with a set of beliefs.

**Definition 3 (AB-POMDP)** *An adversarial-belief POMDP is a tuple* $\mathsf{M} = (S, A, Z, T, O, R, B, \gamma, H)$ *where we define* $S, A, Z, T, O, R, \gamma$ *and* $H$ *as for POMDPs, and* $B \subseteq \Delta(S)$ *is a set of beliefs.*

In an AB-POMDP, the objective is to maximize the expected reward in the POMDP under the worst-case initial belief in $B$. For an AB-POMDP $\mathsf{M}$ and belief $b \in B$, we write $\mathsf{M}_b = (S, A, Z, T, O, R, b, \gamma, H)$ for the POMDP obtained when initializing the AB-POMDP with belief $b$.

**Problem 1** *Given an AB-POMDP* $\mathsf{M}$*, solve* $V_{\mathsf{M}}^* = \max_{\pi \in \Pi_{\mathsf{M}}} \min_{b \in B} V_{\mathsf{M}_b}^\pi$.

When the set of beliefs is the set $\Delta(Q)$ on some subset of states $Q$, any AB-POMDP is equivalent to a zero-sum one-sided POSG, and we codify this result in Theorem 1. In particular, for an AB-POMDP, Theorem 1 gives a recipe to construct a POSG that allows us to find optimal policies for the AB-POMDP. In this POSG, the partially observing player is the agent, and they have the same actions and observations as in the original AB-POMDP. We replace the set of beliefs with a second player whose action set is the set of states $Q$. We shall refer to the partially observing player as the agent, and the fully observing player as nature. By choosing an appropriate distribution over states, nature can choose a distribution in $\Delta(Q)$ against which the agent's policy is evaluated. The optimal policy for the agent in this POSG gives an optimal policy in the original AB-POMDP.

**Theorem 1** *Let* $\mathsf{M} = (S, A, Z, T, O, R, \Delta(Q), \gamma, H)$ *be an AB-POMDP. We define the associated one-sided POSG* $\mathcal{G} = ((S \times \{1, 2\}) \cup \{\bot\}, A, Q, Z \cup \{\top\}, \hat{T}, \hat{O}, \hat{R}, \delta_\bot, \gamma, H + 1)$ *where*

$$\hat{T}(\hat{s}, a, q) = \begin{cases} \delta_{(q,1)} & \hat{s} = \bot, \\ T(s, a) \times \delta_2 & \hat{s} = (s, j), \end{cases} \qquad \hat{O}(\hat{s}, a, q) = \begin{cases} \delta_\top & \hat{s} = \bot \vee \hat{s} = (s, 1), \\ O(s, a) & \hat{s} = (s, 2), \end{cases}$$

*and* $\hat{R}(\hat{s}, a, q) = 0$ *if* $\hat{s} = \bot$ *and* $\hat{R}(\hat{s}, a, q) = \frac{R(s,a)}{\gamma}$ *when* $\hat{s} = (s, j)$, *for all* $\hat{s} \in (S \times \{1, 2\}) \cup \{\bot\}, a \in A$, *and* $q \in Q$. *Additionally, assume that the agent's action set in the POSG at* $\bot$ *is a singleton set* $\{\Diamond\}$ *where* $\Diamond \in A$. *Then, the value of the AB-POMDP* $\mathsf{M}$ *and POSG* $\mathcal{G}$ *are equal, and for any policy* $\sigma \in \Pi_{\mathcal{G}}^1$, *the policy* $\pi$ *in the AB-POMDP given by*

$$\pi(a_1, z_1, \ldots, a_n, z_n) = \sigma(\Diamond, \top, a_1, z_1, \ldots, a_n, z_n)$$

*for* $(a_1, z_1, \ldots, a_n, z_n) \in (A \times Z)^*$, *satisfies* $V_{\mathsf{M}}^\pi = \min_{b \in \Delta(Q)} V_{\mathsf{M}_b}^\pi = V_{\mathcal{G}}^\sigma$.

The proof of Theorem 1 is in Appendix A.1 and follows by establishing mappings between the policy spaces of the AB-POMDP and POSG that preserve value. We add a $\frac{1}{\gamma}$ reward-correction to compensate for the extra step added to the beginning of the game. We expand the state space to ensure the stage-one observation is the dummy observation $\top$. Finally, we restrict the agent's action at $\bot$ so they can not use the initial action as an extra source of randomness to mix over behavioral policies. We can bypass this assumption in finite-horizon settings by applying Kuhn's Theorem [26].

## 4.2 Multi-Environment POMDPs

We now introduce ME-POMDPs and show they are a special case of AB-POMDPs.

**Definition 4 (ME-POMDP)** $\mathcal{M} = (S, A, Z, n, \{T_i\}_{i \in [n]}, \{O_i\}_{i \in [n]}, \{R_i\}_{i \in [n]}, \{b_i\}_{i \in [n]}, \gamma, H)$, *a tuple, is a* multi-environment POMDP *where* $S, A, Z, \gamma$ *and* $H$ *are as in POMDPs , i.e., finite sets of states, actions, and observations, a discount factor, and a horizon. We have* $n \in \mathbb{N}$ *environments and for index* $i \in [n]$, $T_i : S \times A \to \Delta(S)$ *is a transition function,* $O_i : S \times A \to \Delta(Z)$ *is an observation function,* $R_i : S \times A \to \mathbb{R}$ *is a reward function, and* $b_i \in \Delta(S)$ *is an initial state distribution.*

For a fixed $i \in [n]$, the tuple $\mathcal{M}_i = (S, A, Z, n, T_i, O_i, R_i, b_i, \gamma, H)$ defines the $i$-th POMDP in the ME-POMDP. The objective is to maximize the worst-case reward across the environments.

**Problem 2** *Given a ME-POMDP $\mathcal{M}$ solve $V_{\mathcal{M}}^* = \max_{\pi \in \Pi_{\mathcal{M}}} \min_{i \in [n]} V_{\mathcal{M}_i}^{\pi}$.*

In defining a ME-POMDP, we assume that the reward functions $\{R_i\}_{i \in [n]}$ exist on an appropriate scale. For example, if we define a ME-POMDP from expert opinions where one expert uses large rewards to define their environment, the robust policy may be biased toward said environment. One must avoid such cases, for example, by ensuring experts calibrate rewards using the same scale.

We can solve ME-POMDPs using AB-POMDPs. For a ME-POMDP, Theorem 2 gives a recipe to construct an AB-POMDP so that optimal AB-POMDP policies are optimal in the original ME-POMDP. We construct an AB-POMDP where the state space is the product of the original state space and a variable for the environment. The adversary choosing a belief in this AB-POMDP corresponds to the adversary selecting an environment in the ME-POMDP. We formalize this reduction as follows.

**Theorem 2** *For a ME-POMDP $\mathcal{M} = (S, A, Z, n, \{T_i\}_{i \in [n]}, \{O_i\}_{i \in [n]}, \{R_i\}_{i \in [n]}, \{b_i\}_{i \in [n]}, \gamma, H)$, define the associated adversarial-belief POMDP $\hat{\mathsf{M}} = ((S \times [n] \times \{1,2\}) \cup (\{\bot\} \times [n]), A, Z \cup \{\top\}, \hat{T}, \hat{O}, \hat{R}, \Delta(\{\bot\} \times [n]), \gamma, H + 1)$ where for all $\hat{s} \in (S \times [n] \times \{1,2\}) \cup (\{\bot\} \times [n])$ and $a \in A$, we define*

$$\hat{T}(\hat{s}, a) = \begin{cases} b_i \times \delta_i \times \delta_1 & \hat{s} = (\bot, i), \\ T_i(s,a) \times \delta_i \times \delta_2 & \hat{s} = (s, i, j), \end{cases} \quad \hat{O}(\hat{s}, a) = \begin{cases} \delta_{\top} & \hat{s} = (\bot, i) \vee \hat{s} = (s, i, 1), \\ O_i(s,a) & \hat{s} = (s, i, 2), \end{cases}$$

*and $\hat{R}(\hat{s}, a) = 0$ if $\hat{s} = (\bot, i)$, and $\hat{R}(\hat{s}, a) = {R_i(s,a)}/{\gamma}$ when $\hat{s} = (s, i, j)$. Additionally, assume that the agent's action set in the AB-POMDP at states in $\{\bot\} \times [n]$ is a singleton set $\{\Diamond\}$ where $\Diamond \in A$. Then, for any policy $\sigma \in \Pi_{\hat{\mathsf{M}}}$, the policy $\pi$ in the ME-POMDP given by*

$$\pi(a_1, z_1, \ldots, a_n, z_n) = \sigma(\Diamond, \top, a_1, z_1, \ldots, a_n, z_n) \quad \forall (a_1, z_1, \ldots, a_n, z_n) \in (A \times Z)^*$$

*satisfies $\min_{i \in [n]} V_{\mathcal{M}_i}^{\pi} = \min_{b \in \Delta(\{\bot\} \times [n])} V_{\hat{\mathsf{M}}_b}^{\sigma}$. Also, the values are equal, i.e., $V_{\mathcal{M}}^* = V_{\hat{\mathsf{M}}}^*$.*

The proof of Theorem 2 is nearly identical to that of Theorem 1, as we elaborate in Appendix A.1.

### 4.3 Restricted Models and Reductions

ME-POMDPs may differ in their transition, observation, and reward functions. By requiring all environments to either share a transition or observation function, we get restricted models. When the observation function $O_i$ does not change with the environment, we label the model as a *partially observable multi-environment MDP* (PO-MEMDP), *i.e.*, a multi-environment MDP (MEMDP) extended with an observation function. PO-MEMDPs are equivalent to hidden-model POMDPs [15].

We can transform an AB-POMDP where the belief set $B$ is the set of distributions over a state subset into a PO-MEMDP, while preserving optimal policies, and Theorem 3 encodes this transformation. Given $B = \Delta(Q)$ for a state subset $Q$ and a policy $\pi$, the worst-case belief is $\delta_q$ for some $q \in Q$, and defines an initial state. The PO-MEMDP encodes these possible initial states.

**Theorem 3** *Given an AB-POMDP $\mathsf{M} = (S, A, Z, T, O, R, \Delta(Q), \gamma, H)$ where the belief set is $\Delta(Q)$ for a set of states $Q \subseteq S$, define an associated PO-MEMDP $\hat{\mathcal{M}} = ((S \times \{1,2\}) \cup \{\bot\}, A, Z \cup \{\top\}, |Q|, \{\hat{T}_q\}_{q \in Q}, \hat{O}, \hat{R}, \delta_{\bot}, \gamma, H + 1)$, where $\hat{T}$, $\hat{O}$ and $\hat{R}$ are as follows.*

$$\hat{T}_q(\hat{s}, a) = \begin{cases} \delta_q \times \delta_1 & \hat{s} = \bot, \\ T(s,a) \times \delta_2 & \hat{s} = (s, j), \end{cases} \quad \hat{O}(\hat{s}, a) = \begin{cases} \delta_{\top} & \hat{s} = \bot \vee \hat{s} = (s, 1), \\ O(s,a) & \hat{s} = (s, 2). \end{cases}$$

*Meanwhile, $\hat{R}(\hat{s}, a) = {R(s,a)}/{\gamma}$ if $\hat{s} = (s, j)$ for some $s \in S, j \in \{1,2\}$ and $\hat{R}(\bot, a) = 0$. Also, assume that the agent's action set in the PO-MEMDP at $\bot$ is a singleton set $\{\Diamond\}$ where $\Diamond \in A$. Then, for any policy $\sigma \in \Pi_{\hat{\mathcal{M}}}$, the policy $\pi$ in the AB-POMDP given by*

$$\pi(a_1, z_1, \ldots, a_n, z_n) = \sigma(\Diamond, \top, a_1, z_1, \ldots, a_n, z_n) \quad \forall (a_1, z_1, \ldots, a_n, z_n) \in (A \times Z)^*$$

*satisfies $\min_{b \in \Delta(Q)} V_{\mathsf{M}_b}^{\pi} = \min_{q \in Q} V_{\hat{\mathcal{M}}_q}^{\sigma}$, and the values are equal, that is, $V_{\hat{\mathcal{M}}}^* = V_{\mathsf{M}}^*$.*

The proof of Theorem 3 again follows the same techniques as Theorem 1 as we show in Appendix A.1. Note that we slightly abuse notation by indexing ME-POMDP models with states $q \in Q$.

Theorem 3 shows that AB-POMDPs are equivalent to ME-POMDPs as PO-MEMDPs are a subset of ME-POMDPs. Additionally, Theorem 3 shows that we can represent any ME-POMDP with a polynomial larger model with multiple transition functions, *i.e.*, a PO-MEMDP.

When the transitions $T$ and initial distribution $b$ do not change across the environments, *i.e.*, $(T_i, b_i) = (T_j, b_j)$ for all $i, j \in [n]$, we refer to the model as a *multi-observation POMDP* (MO-POMDP). By Theorem 4, for any PO-MEMDP, we can construct a MO-POMDP with the same optimal policy.

**Theorem 4** *Given a PO-MEMDP* $\mathcal{M} = (S, A, Z, [n], \{T_i\}_{i \in [n]}, O, \{R_i\}_{i \in [n]}, \{b_i\}_{i \in [n]}, \gamma, H)$, *define a MO-POMDP* $\hat{\mathcal{M}} = (S^{[n]}, A, Z, [n], \hat{T}, \{\hat{O}_i\}_{i \in [n]}, \{\hat{R}_i\}_{i \in [n]}, b_1 \times \cdots \times b_n, \gamma, H)$ *such that* $\hat{T}(s_1, \ldots, s_n, a) = T_1(s_1, a) \times \cdots \times T_n(s_n, a)$, $\hat{O}_i(s_1, \ldots, s_n, a) = O(s_i, a)$, *and* $\hat{R}_i(s_1, \ldots, s_n, a) = R_i(s_i, a)$. *The policy sets satisfy* $\Pi_{\mathcal{M}} = \Pi_{\hat{\mathcal{M}}}$, *and for all* $\pi \in \Pi_{\mathcal{M}}$, *we have* $\min_{i \in [n]} V_{\mathcal{M}_i}^{\pi} = \min_{i \in [n]} V_{\hat{\mathcal{M}}_i}^{\pi}$.

The resulting MO-POMDP simulates all environments in the state space. Changing the environment changes the copy of the state that generates observations and rewards, and thus, for a policy $\pi$ and environment $i$, the two models have the same reward. The full proof of Theorem 4 is in Appendix A.1.

Theorem 4 requires multiple reward functions in the MO-POMDP, and we prove that these are, in fact, necessary for the finite-horizon case in Appendix A.2. An illustrative example of the utility of ME-POMDPs, MO-POMDPs, and PO-MEMDPs, adapted from [8], can be found in Appendix B.

## 5 Algorithms for AB-POMDPs

We provide algorithms to solve AB-POMDPs by combining value iteration and linear programming. AB-POMDPs are equal to POMDPs up to how we specify the initial belief. We show that given a piecewise-linear convex value function for the POMDP, we can compute the value of the AB-POMDP by minimizing the value function. This problem is a linear program (LP). Additionally, we can use the dual LP solution to construct a policy for the agent that attains the value.

### 5.1 Computing Policies by Solving Linear Programs

When the value function has a piecewise-linear convex representation through a set $\Gamma$ of $\alpha$-vectors, and the belief set $B$ is of the form $\Delta(Q)$ for some subset of states $Q \subseteq S$, we can minimize the value function by solving a linear program. Indeed, we minimize the value function for beliefs in $B$ by solving $\min_{b \in \Delta(Q)} \max_{\alpha \in \Gamma} \alpha \cdot b$, and this problem can be expressed in the LP in (3).

The resulting value $v$ is the value that nature guarantees by playing belief $b$, *i.e.*, no matter the policy the agent plays, if nature plays $b$, the reward will not be greater than $v$. Minimizing the value function upper bounds the AB-POMDP's value, but it remains to show that a policy exists attaining this bound.

We construct a policy for the agent that attains this value by solving the dual LP, presented in (4). Each $\alpha$-vector corresponds to a deterministic history-dependent policy. In the policy corresponding to the solution $y$ to (4), the agent draws an $\alpha$-vector according to $y$ and plays the corresponding history-dependent policy. The resulting value $v$ is the value that the agent can guarantee by playing the policy corresponding to $y$, *i.e.*, no matter the initial belief nature plays, if the agent plays $y$, the reward will not be less than $v$. In Theorem 5, we show that the policy we construct from the solution to the LP in (4) is optimal.

**Theorem 5** *Let* $\mathsf{M} = (S, A, Z, T, O, R, \Delta(Q), \gamma, H)$ *be an AB-POMDP, let* $\Gamma$ *be a finite set of* $\alpha$-vectors such that $\max_{\alpha \in \Gamma} \alpha \cdot b$ is the value function for this AB-POMDP, and let $\mathsf{Pol} \colon \Gamma \to \Pi_{\mathsf{M}}$

$$
\begin{aligned}
\min_{b \in \mathbb{R}^Q, v \in \mathbb{R}} \quad & v, & (3) \\
\text{s.t.} \quad & \forall \alpha \in \Gamma : \sum_{s \in Q} \alpha(s)b(s) \leq v, \\
& \forall s \in Q : b(s) \geq 0, \\
& \sum_{s \in Q} b(s) = 1.
\end{aligned}
$$

$$
\begin{aligned}
\max_{y \in \mathbb{R}^\Gamma, v \in \mathbb{R}} \quad & v, & (4) \\
\text{s.t.} \quad & \forall s \in Q : \sum_{\alpha \in \Gamma} \alpha(s)y(\alpha) \geq v, \\
& \forall \alpha \in \Gamma : y(\alpha) \geq 0, \\
& \sum_{\alpha \in \Gamma} y(\alpha) = 1.
\end{aligned}
$$

*be a mapping that returns a deterministic history-dependent policy $\pi$ such that $V_{\mathsf{M}_b}^{\mathsf{Pol}(\alpha)} = \alpha \cdot b$. If $y \in \mathbb{R}^\Gamma$ is the solution to LP (4), then the policy for the agent where they draw an $\alpha$-vector randomly according to $y$ and play the corresponding history-dependent policy is an optimal policy for $\mathsf{M}$.*

The result above shows that solving an adversarial-belief POMDP reduces to solving a zero-sum game where nature plays beliefs and the agent plays $\alpha$-vectors. Indeed, the LPs above encode the problem of solving a static zero-sum game [3]. Both LPs correspond to minimizing a piecewise-linear convex function over a compact set, so solutions exist. Theorem 5's proof simply applies the definition of the LPs, and we detail this proof in Appendix A.3.

We remark that while Theorem 5 describes a procedure to construct a mixed policy, that is, a mixture of deterministic policies, we can construct a behavioral policy in $\Pi_{\mathsf{M}}$ with the same value, following Kuhn's theorem [26], and we detail this construction in Appendix A.4.

We additionally remark that if $\max_{\alpha \in \Gamma} \alpha \cdot b$ under-approximates the value function, we can still construct a policy that attains the value $\min_{b \in \Delta(Q)} \max_{\alpha \in \Gamma} \alpha \cdot b$, as long as Pol exists and satisfies $V_{\mathsf{M}_b}^{\mathsf{Pol}(\alpha)} \geq \alpha \cdot b$. We discuss implementing the Pol mapping in Appendix A.5.

### 5.2 Adversarial-Belief HSVI

Since AB-POMDPs are equal to POMDPs up to the initial belief, we can use well-known POMDP methods to generate the $\alpha$-vectors that form (an approximation of) the value function. For example, for a finite-horizon $H$, we can compute $\Gamma_H$ according to Equations (1) and (2). $\Gamma_H$ represents all deterministic history-based policies of length $H$ and does not require specifying the initial state distribution. We can, there-

---

**Algorithm 1** AB-HSVI

**Input:** $\gamma \in [0, 1), \epsilon > 0$
Initialize $\Upsilon$ with Fast Informed Bound
Initialize $\Gamma$ with "always play action $a$" $\alpha$-vector $\forall a \in A$
$b \leftarrow$ worst-case state distribution in $\Gamma$ using LP (3)
**while** Gap$(\Upsilon, \Gamma, b) \geq \epsilon$ **do**
    $\Upsilon, \Gamma \leftarrow$ one iteration of HSVI$(\Upsilon, \Gamma, b, \gamma, \epsilon)$
    $b \leftarrow$ worst-case state distribution in $\Gamma$ using LP (3)
**end while**

---

fore, compute the optimal value and robust agent and nature policies by applying the LPs (3) and (4) to $\Gamma_H$. Note that we can prune dominated $\alpha$-vectors in $\Gamma_H$ without influencing the result of the LPs.

To construct a more efficient algorithm, we can generate $\alpha$-vectors that approximate the optimal value function in the infinite-horizon setting using approximate $\alpha$-vector-based POMDP methods such as HSVI. As explained in Section 3, HSVI provably converges to a gap between upper and lower bounds on the optimal value function of less than a predefined $\epsilon$ at a given initial state distribution. If we use an arbitrary initial state distribution and run HSVI as-is, the algorithm converges for that state distribution, but there are no guarantees for the upper-lower-bound gap at other distributions. However, the lower bound is still a sound under-approximation of the value function.

We use this observation to construct a more sophisticated solution, which we call *adversarial-belief HSVI* (AB-HSVI, Algorithm 1). We compute the worst-case initial state distribution between each depth-first search using LP (3), and start the next depth-first search from this distribution. Essentially, this procedure restarts HSVI, initializing with the upper and lower bounds of the previous iteration. This algorithm terminates once the worst-case initial state distribution has a gap between the upper and lower bounds of less than $\epsilon$, giving us a tighter approximation.

## 6 Experimental Evaluation

The implementation of the LPs (3) and (4) along with AB-HSVI (Algorithm 1) forms a solution method for ME-POMDPs, and we answer the following research questions regarding this method.

    **(Q1) Scalability**: What is the computational cost of solving AB-POMDPs?
    **(Q2) Baseline comparison**: What is the added difficulty of robustness against adversarial beliefs compared to a naive baseline of solving individual POMDPs?
    **(Q3) Model formulation**: Does the model type, *i.e.*, whether the problem is formulated as a ME-POMDP, PO-MEMDP, MO-POMDP or AB-POMDP, influence the performance?

As no benchmarks exist for ME-POMDPs, we introduce two benchmarks for our experimental evaluation. The first benchmark is based on the endangered bird preservation case study presented in

Table 1: Lower bound value, time of convergence, and left-over gap between upper and lower bound of the Bird problem for various problem sizes and model types.

| Model | Properties | | | | PO-MEMDP | | | MO-POMDP | | | ME-POMDP | | |
|---|---|---|---|---|---|---|---|---|---|---|---|---|---|
| | $|S|$ | $n$ | $|A|$ | $|Z|$ | $V_{<tl}$ | Conv (s) | Gap | $V_{<tl}$ | Conv (s) | Gap | $V_{<tl}$ | Conv (s) | Gap |
| $BP_{3,3,3}$ | 3 | 3 | 3 | 2 | 68.26 | 58.50 | $< \epsilon$ | 70.44 | 84.98 | $< \epsilon$ | 69.62 | 2039.31 | $< \epsilon$ |
| $BP_{3,3,4}$ | 3 | 3 | 4 | 2 | 44.44 | - | 4.33 | 54.85 | 2976.33 | $< \epsilon$ | 44.79 | - | 6.02 |
| $BP_{3,3,5}$ | 3 | 3 | 5 | 2 | 74.58 | 3104.30 | $< \epsilon$ | 80.01 | 21.08 | $< \epsilon$ | 74.59 | - | 0.61 |
| $BP_{3,4,3}$ | 3 | 4 | 3 | 2 | 20.48 | - | 7.80 | 24.09 | 118.82 | $< \epsilon$ | 22.56 | - | 5.81 |
| $BP_{3,5,3}$ | 3 | 5 | 3 | 2 | 31.23 | - | 11.63 | 31.85 | 175.99 | $< \epsilon$ | 32.73 | - | 9.74 |
| $BP_{4,3,3}$ | 4 | 3 | 3 | 2 | 63.91 | - | 19.57 | 73.49 | - | 2.51 | 55.96 | - | 28.56 |
| $BP_{5,3,3}$ | 5 | 3 | 3 | 2 | 35.57 | - | 6.76 | 36.04 | - | 5.30 | 35.84 | - | 6.77 |

Appendix B, which we shall refer to as the *Bird problem*. We extend the model to ME-POMDPs, PO-MEMDPs, and MO-POMDPs, using randomization to generate transition and observation functions to obtain non-trivial problem instances. In particular, we parameterize the number of states $|S| \geq 2$, actions $|A|$, and experts $n$. We denote instances of this benchmark as $BP_{|S|,|A|,n}$.

**Remark 1** *We exclusively use randomization to create challenging ME-POMDP problem instances. Even with randomization, creating challenging environments is difficult. When generating* 100 *random models for the Bird problems with* 3 *states,* 3 *actions, and* 3 *experts, we only found* 35 *out of* 100 *non-trivial PO-MEMDPs, where a model is trivial if we can solve it in less than* 30 *seconds.*

For the second benchmark, we extend *RockSample* [40] to ME-POMDPs. We parameterize the grid size $m$, good rocks $g$, and total number of rocks $t$, and denote instances of this benchmark as $RS_{m,g,t}$. We consider randomized and relatively fixed rock positions. We denote the RockSample instances with fixed rock positions as $RS^c_{m,g,t}$. See Appendix C for full details on the benchmarks construction.

We set a time limit $tl$ of 3600 seconds, discount factor $\gamma = 0.95$, and set HSVI's gap threshold to $\epsilon = 0.1 \cdot R_{\min}$ where $R_{\min}$ is the minimum problem reward. We use sparse matrices and prune fully dominated $\alpha$-vectors. We run experiments on a computer with an Intel Core i9-10980XE 3.00GHz processor and 256GB of RAM. We use Gurobi [18] to solve LPs. All code is available at [6].

**Results and Discussion**

**(Q1) Scalability** Tables 1 and 2 show the results of running AB-HSVI on the Bird problem and RockSample. In both problems, the convergence times and gaps increase with the number of environments. The RockSample problems generally converge faster than Bird problems, likely due to RockSample's terminal state. We note that the structure of the environments has a great effect on the difficulty of the problems. In Figure 2, we show that the relative positions of the rocks, *i.e.*, whether they are close or far to the agent's initial position, have a significant influence on AB-HSVI's convergence time. The relationship between environment configuration and solve time explains why, for the Bird problem, the convergence times and gaps are not monotonic in the problem size.

**(Q2) Baseline comparison** We compare AB-HSVI with the values and time required to solve all individual environments (*i.e.*, standard POMDPs) on RockSample. We summarize the results in Figure 4 and give details in Appendix D. The time increase for the ME-POMDP computation, shown in Table 3, primarily scales with the number of environments. We also note that robust values achieve an expected reward that is close to the rewards in individual models, and the robust value far exceeds the worst case of playing the optimal policy for an incorrectly assumed environment.

**(Q3) Model formulation** For a Bird problem ME-POMDP, Table 1 shows how solve time and value vary when we either (1) fix observation functions to get a PO-MEMDP, or (2) fix transitions to get a MO-POMDP. AB-HSVI tends to converge more quickly and return higher values for MO-POMDPs, showing that uncertain observation functions are easier to handle than uncertain transitions.

We can formulate problems as either AB-POMDPs or ME-POMDPs, and we compare these formulations for RockSample in Figure 3. In all but two instances, AB-POMDPs converge faster than ME-POMDPs. Also, gaps between convergence times increase with the number of environments. Finally, we note that AB-POMDPs report slightly higher values than ME-POMDPs, but the difference is less than the error $\epsilon$. Details on the two formulations and the results are in Appendices C and D.

Table 2: Lower bound value, time of convergence, and left-over gap between upper and lower bound of the RockSample problem for various problem sizes with rocks nearby or far away.

| Model | Properties | | | | Rocks nearby | | | Rocks far away | | |
|---|---|---|---|---|---|---|---|---|---|---|
| | $\|S\|$ | $n$ | $\|A\|$ | $\|Z\|$ | $V_{<tl}$ | Conv (s) | Gap | $V_{<tl}$ | Conv (s) | Gap |
| $RS^c_{2,1,2}$ | 9 | 2 | 7 | 3 | 16.53 | 11.70 | $< \epsilon$ | 16.53 | 11.70 | $< \epsilon$ |
| $RS^c_{3,1,2}$ | 19 | 2 | 7 | 3 | 16.14 | 52.74 | $< \epsilon$ | 14.68 | 169.95 | $< \epsilon$ |
| $RS^c_{4,1,2}$ | 33 | 2 | 7 | 3 | 15.48 | 130.77 | $< \epsilon$ | 13.02 | 1588.97 | $< \epsilon$ |
| $RS^c_{5,1,2}$ | 51 | 2 | 7 | 3 | 15.40 | 331.37 | $< \epsilon$ | 11.03 | - | 1.46 |
| $RS^c_{6,1,2}$ | 73 | 2 | 7 | 3 | 14.52 | 640.40 | $< \epsilon$ | | | |
| $RS^c_{7,1,2}$ | 99 | 2 | 7 | 3 | 14.54 | 1280.66 | $< \epsilon$ | | | |
| $RS^c_{2,1,3}$ | 9 | 3 | 8 | 3 | 15.90 | 115.11 | $< \epsilon$ | 15.90 | 115.11 | $< \epsilon$ |
| $RS^c_{3,1,3}$ | 19 | 3 | 8 | 3 | 15.41 | 269.10 | $< \epsilon$ | 14.34 | 1072.32 | $< \epsilon$ |
| $RS^c_{4,1,3}$ | 33 | 3 | 8 | 3 | 15.14 | 787.82 | $< \epsilon$ | 11.11 | - | 2.73 |
| $RS^c_{5,1,3}$ | 51 | 3 | 8 | 3 | 14.80 | 1793.75 | $< \epsilon$ | 8.15 | - | 5.34 |
| $RS^c_{6,1,3}$ | 73 | 3 | 8 | 3 | 14.31 | 2556.11 | $< \epsilon$ | | | |
| $RS^c_{7,1,3}$ | 99 | 3 | 8 | 3 | 13.30 | - | 2.25 | | | |

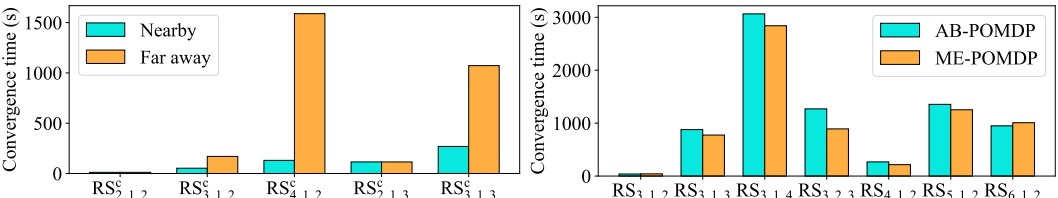

Figure 2: Convergence time of RockSample instances with rocks nearby vs. far away.

Figure 3: Convergence time of RockSample problems modeled as AB-POMDPs vs. ME-POMDPs.

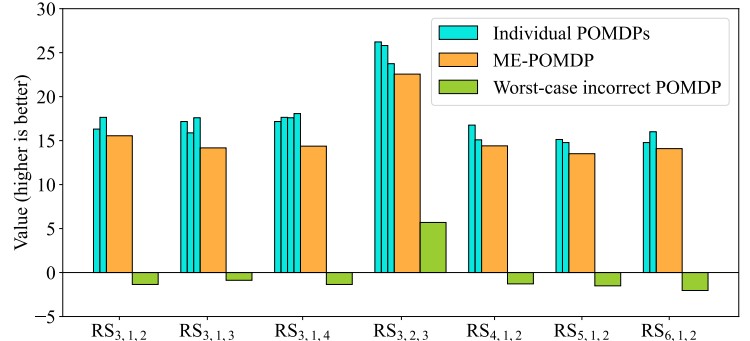

Figure 4: Lower bound values of POMDPs in a ME-POMDP, the ME-POMDP, and worst-case missassumed POMDP for RockSample instances.

Table 3: Convergence time increase from summed individual POMDPs to ME-POMDP in Figure 4.

| Model | Factor |
|---|---|
| $RS_{3,1,2}$ | 2.11 |
| $RS_{3,1,3}$ | 17.68 |
| $RS_{3,1,4}$ | 65.09 |
| $RS_{3,2,3}$ | 5.49 |
| $RS_{4,1,2}$ | 2.15 |
| $RS_{5,1,2}$ | 4.84 |
| $RS_{6,1,2}$ | 2.50 |

## 7 Conclusion

We presented new results on multi-environment POMDPs, *i.e.*, discrete sets of POMDPs for which we need to compute a single policy that maximizes the worst-case expected reward. We introduced adversarial-belief POMDPs as an overarching model and showed how these AB-POMDPs are a special case of partially observable stochastic games. Leveraging the understanding of ME-POMDPs as AB-POMDPs, we developed exact and point-based algorithms for computing policies in ME-POMDPs. Future work will investigate more efficient algorithms by leveraging the structure in ME-POMDPs and AB-POMDPs, or by using additional HSVI optimization techniques such as tracking previously explored beliefs and using compact state space representations [33].

**Limitations** The main limitation of this work is the scalability of AB-HSVI, particularly, the substantial increase in convergence time with the number of environments. We believe that exploring policy-gradient or online-planning methods for ME-POMDPs is a critical next step to ensuring their applicability, and we believe that our theoretical results provide a foundation for this work.

# 8   Acknowledgements

We would like to thank the anonymous reviewers for their useful comments. This work has been partially funded by the ERC Starting Grant DEUCE (101077178). This work has also been partially supported by the Air Force Office of Scientific Research (AFOSR) under grant number FA9550-22-1-0403, and by the Office of Naval Research (ONR) under grant number N00014-24-1-2797. It has also been supported by the FWO "SynthEx" project (G0AH524N).

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

# A  Complete Proofs

## A.1  Proofs from Section 4

**Proof of Theorem 1.** We prove Theorem 1 by constructing mappings between the policy spaces for the adversarial-belief POMDP (AB-POMDP) M and the partially observable stochastic game (POSG) $\mathcal{G}$ that preserve value.

We first recall the policy spaces for M and $\mathcal{G}$. Let $\Pi_{\mathsf{M}}$ define the policy space for the AB-POMDP M, and let $\Pi_{\mathcal{G}}^1$ be the policy space for the agent in the POSG $\mathcal{G}$. Recall that a policy $\pi \in \Pi_{\mathsf{M}}$ is a mapping

$$\pi \colon (A \times Z)^* \to \Delta(A), \tag{5}$$

while a policy $\sigma \in \Pi_{\mathcal{G}}^1$ is a mapping

$$\sigma \colon (A \times (Z \cup \{\top\}))^* \to \Delta(A). \tag{6}$$

We note that due to the restriction on the agent's actions at the initial state, the initial action given by $\sigma$ is deterministic and always evaluates to $\Diamond$. That is, $\sigma(\epsilon) = \Diamond$, where $\epsilon$ is the empty history.

We claim that for any policy in the AB-POMDP, we can construct a corresponding policy in the POSG with the same reward, and vice versa.

To prove the first direction in the above statement, define a mapping $f \colon \Pi_{\mathsf{M}} \to \Pi_{\mathcal{G}}^1$ as follows

$$f(\pi)(a_1, z_1, a_2, z_2, \ldots, a_n, z_n) = \begin{cases} \pi(a_2, z_2, \ldots, a_n, z_n) & n \geq 2, \wedge z_2 \neq \top \wedge \ldots \wedge z_n \neq \top, \\ \pi(\epsilon) & n = 1, \\ \delta_{\Diamond} & \text{otherwise.} \end{cases} \tag{7}$$

We note that this definition implies $f(\pi)(\epsilon) = \delta_{\Diamond}$, and thus $f(\pi)$ is consistent with $\mathcal{G}$. The first two cases define the policies for feasible trajectories in the game, as only the first observation will be $\top$. The last case handles the initial action, and ensures the policy is well-defined.

We claim that $\pi$ and $f(\pi)$ have the same value. That is,

$$\min_{b \in \Delta(Q)} V_{\mathsf{M}_b}^{\pi} = \min_{\pi_2 \in \Pi_{\mathcal{G}}^2} V_{\mathcal{G}}^{f(\pi), \pi_2}, \tag{8}$$

where we use $V_{\mathcal{G}}^{\pi_1, \pi_2}$ to denote the agent's expected reward in the game $\mathcal{G}$ when the agent and nature play policies $\pi_1$ and $\pi_2$ respectively. This fact follows from the structure of the game. Indeed, the structure of the game $\mathcal{G}$ is such that at the initial state $\bot$, nature's action defines an initial state in $S \times \{1, 2\}$. States in $S \times \{1, 2\}$ are then closed under transitions in the game $\mathcal{G}$, and have identical dynamics to M up to the flag in $\{1, 2\}$, with nature's actions having no effect. For completeness, we carry out this reasoning in its entirety below.

We first compute $V_{\mathcal{G}}^{f(\pi), \pi_2}|_{\hat{s}_2 = (q, 1)}$, where we let $V_{\mathcal{G}}^{f(\pi), \pi_2}|_{\hat{s}_2 = (q, 1)}$ denote the reward that the agent receives from stage 2 onward when $\hat{s}_2 = (q, 1)$ for some $q \in S$. We use $\hat{s}_t$ to denote the state in the game $\mathcal{G}$ at time $t$, and for all $t \geq 2$, we note that by definition of $\mathcal{G}$, $\hat{s}_t$ will be of the form $(s_t, i_t)$ for some $s_t \in S$ and $i_t \in \{1, 2\}$.

We can apply the definition of the game $\mathcal{G}$ and the policy $f(\pi)$ to obtain

$$V_{\mathcal{G}}^{f(\pi), \pi_2}|_{\hat{s}_2 = (q, 1)} = \mathbb{E}\left[ \sum_{t=2}^{H+1} \gamma^{t-1} \hat{R}(\hat{s}_t, a_t) | \hat{s}_2 = (q, 1) \right] \tag{9}$$

where

$$\hat{s}_{t+1} \sim \hat{T}(\hat{s}_t, a_t, r_t) \qquad \forall t \geq 2, \tag{10}$$
$$a_{t+1} \sim f(\pi)(a_1, z_1, \ldots, a_t, z_t) \qquad \forall t \geq 2, \tag{11}$$
$$r_{t+1} \sim \pi_2(h_t) \qquad \forall t \geq 2, \tag{12}$$
$$z_t \sim \hat{O}(\hat{s}_{t+1}, a_t, r_t) \qquad \forall t \geq 2. \tag{13}$$

For each $t \geq 3$, we have $\hat{s}_t = (s_t, 2)$ for some $s_t \in S$, by definition of $\hat{T}$, and for all $t \geq 2$, we have $s_{t+1} \sim T(s_t, a_t)$, again by definition of $\hat{T}$. By assumption, $s_2 = q$. We then note that, by definition

of $\mathcal{G}$, and the fact that $\hat{s}_2 \in S \times \{1\}$, that $z_1 = \top$, and by definition of $f(\pi)$ we also have $a_1 = \Diamond$, and hence we have $f(\pi)(a_1, z_1, a_2, z_2 \dots, a_t, z_t) = \pi(a_2, z_2, \dots, a_t, z_t)$, and $f(\pi)(a_1, z_1) = \pi(\epsilon)$. For $t \geq 2$, we have $z_t \sim \hat{O}(\hat{s}_{t+1}, a_t) = \hat{O}((s_{t+1}, 2), a_t) = O(s_{t+1}, a)$. Finally, expanding the reward, we have $\hat{R}(\hat{s}_t, a_t) = R(s_t, a_t)/\gamma$ for all $t \geq 2$. Thus, we can rewrite the expectation as

$$= \mathbb{E}\left[\sum_{t=2}^{H+1} \gamma^{t-2} R(s_t, a_t) | s_2 = q\right] \tag{14}$$

where

$$s_{t+1} \sim T(s_t, a_t) \qquad\qquad \forall t \geq 2, \tag{15}$$
$$a_2 \sim \pi(\epsilon), \tag{16}$$
$$a_{t+1} \sim \pi(a_2, z_2, \dots, a_t, z_t) \qquad\qquad \forall t \geq 2, \tag{17}$$
$$z_t \sim O(s_{t+1}, a_t) \qquad\qquad \forall t \geq 2. \tag{18}$$

However, up to relabeling states and timesteps such that $s'_t = s_{t+1}$, $a'_t = a_{t+1}$ and $z'_t = z_{t+1}$, this expectation is exactly

$$\mathbb{E}\left[\sum_{t=1}^{H} \gamma^{t-1} R(s'_t, a'_t) | s'_1 = q\right], \tag{19}$$

where

$$s'_{t+1} \sim T(s'_t, a_t) \qquad\qquad \forall t \geq 1, \tag{20}$$
$$a'_1 \sim \pi(\epsilon), \tag{21}$$
$$a'_{t+1} \sim \pi(a'_1, z'_1, \dots, a'_t, z'_t) \qquad\qquad \forall t \geq 1, \tag{22}$$
$$z'_t \sim O(s'_{t+1}, a'_t) \qquad\qquad \forall t \geq 1. \tag{23}$$

This expectation is then equal to $V^\pi_{\mathsf{M}_{\delta_q}}$, and thus we deduce

$$V^{f(\pi), \pi_2}_{\mathcal{G}}|_{\hat{s}_2 = (q,1)} = V^\pi_{\mathsf{M}_{\delta_q}}. \tag{24}$$

We now compute the agent's total reward in $\mathcal{G}$. By (24), and the fact that the reward from the first stage is always $0$ for the agent, we have

$$V^{f(\pi), \pi_2}_{\mathcal{G}} = \sum_{q \in Q} \mathbb{P}\left[\hat{s}_2 = (q,1)\right] \cdot V^{f(\pi), \pi_2}_{\mathcal{G}}|_{\hat{s}_2 = (q,1)}, \tag{25}$$

and by the definition of the game $\mathcal{G}$, we have $\mathbb{P}\left[\hat{s}_2 = (q,1)\right] = \mathbb{P}[\pi_2(\bot) = q]$.

Finally, to prove that (8) holds, we construct value-preserving mappings from $\Delta(Q)$ to $\Pi^2_{\mathcal{G}}$ and vice versa. This fact holds as the agent's reward only depends on nature's policy through the actions at $\bot$, and so each belief is associated with a class of policies that have that belief as the initial action distribution for the nature player. Define, for a given belief $b \in \Delta(Q)$, a policy $\pi^{(b)}_2 \in \Pi^2_{\mathcal{G}}$ with

$$\pi^{(b)}_2(h) = \begin{cases} b & h = \bot, \\ \rho & \text{otherwise}, \end{cases} \tag{26}$$

where $\rho \in Q$ is arbitrary. We have, by definition of $\pi^{(b)}_2$, that

$$V^{f(\pi), \pi^{(b)}_2}_{\mathcal{G}} = \sum_{q \in Q} b(q) \cdot V^{f(\pi), \pi^{(b)}_2}_{\mathcal{G}}|_{\hat{s}_2 = (q,1)} = \sum_{q \in Q} b(q) \cdot V^\pi_{\mathsf{M}_{\delta_q}} = V^\pi_{\mathsf{M}_b}. \tag{27}$$

Thus, for any belief $b$, there exists a policy for nature that attains the same value in the POSG as the belief $b$ would induce in the AB-POMDP. Similarly, given a policy $\pi_2 \in \Pi^2_{\mathcal{G}}$, if we define $b^{\pi_2} = \pi_2(\bot)$, we have

$$V^\pi_{\mathsf{M}_{(b^{\pi_2})}} = \sum_{q \in Q} b^{\pi_2}(q) \cdot V^\pi_{\mathsf{M}_{\delta_q}} = \sum_{q \in Q} \mathbb{P}\left[\pi_2(\bot) = q\right] V^{f(\pi), \pi_2}_{\mathcal{G}}|_{\hat{s}_2 = (q,1)} = V^{f(\pi), \pi_2}_{\mathcal{G}}. \tag{28}$$

With this equality, we can conclude (8) holds.

We have proven that for any agent policy in M, we can construct an agent policy $f(\pi)$ in $\mathcal{G}$ with the same value. This fact establishes one direction of the equality of the value of M and $\mathcal{G}$, that is

$$\max_{\sigma \in \Pi_{\mathcal{G}}^1} V_{\mathcal{G}}^{\sigma} \geq \max_{\pi \in \Pi_{\mathsf{M}}} V_{\mathsf{M}}^{\pi}. \tag{29}$$

Mapping $f$ is not a bijection, but for each POSG policy $\sigma \in \Pi_{\mathcal{G}}^1$ we can construct a policy $\sigma' \in \Pi_{\mathcal{G}}^1$ such that $\sigma' \in \mathsf{Range}(f)$, and $\sigma'$ has the same value as $\sigma$, and proving this fact will complete the proof.

Given $\sigma$, define $q_{\sigma} \in \Pi_{\mathsf{M}}$ such that

$$q_{\sigma}(a_1, z_1, \ldots, a_n, z_n) = \sigma(\Diamond, \top, a_1, z_1, \ldots, a_n, z_n) \quad \forall (a_1, z_1, \ldots, a_n, z_n) \in (A \times Z)^*. \tag{30}$$

Note that we, by an abuse of notation, use this definition to communicate that $q_{\sigma}(\epsilon) = \sigma(\Diamond, \top)$. We can then write $\sigma$ as

$$\sigma(\epsilon) = \delta_{\Diamond}, \tag{31}$$
$$\sigma(\Diamond, \top) = q_{\sigma}(\epsilon), \tag{32}$$

$$\sigma(\Diamond, \top, a_2, z_2, \ldots, a_n, z_n) = \begin{cases} q_{\sigma}(a_2, z_2, \ldots, a_n, z_n) & z_2 \neq \top \wedge \ldots \wedge z_n \neq \top, \\ r(a_2, z_2, \ldots, a_n, z_n) & \text{otherwise,} \end{cases} \tag{33}$$

$$\sigma(a_1, z_1, \ldots, a_n, z_n) = w(a_1, z_1, \ldots, a_n, z_n), \quad \forall a_1 \neq \Diamond, z_1 \neq \top, \tag{34}$$

where $r$ and $w$ are some mappings from $(A \times (Z \cup \{\top\}))^*$ to $\Delta(A)$ determined by $\sigma$. By (8) we have that $V_{\mathcal{G}}^{f(q_{\sigma})} = V_{\mathsf{M}}^{q_{\sigma}}$. However, by definition of $\mathcal{G}$, $a_1 = \Diamond$, $z_1 = \top$, and $z_i \in Z$ for all $i \geq 2$. Hence, $\sigma$ and $f(q_{\sigma})$ will agree for any feasible path. Thus, we conclude that in fact

$$V_{\mathsf{M}}^{q_{\sigma}} = V_{\mathcal{G}}^{f(q_{\sigma})} = V_{\mathcal{G}}^{\sigma}. \tag{35}$$

We can now deduce that, for any POSG policy, there exists an AB-POMDP policy with the same value, thus establishing the other direction in the value equality between $\mathcal{G}$ and M. Indeed, the policy $f(q_{\sigma})$ is the required $\sigma'$ which has the same value as $\sigma$, but lies in the range of $f$. Additionally, if we set $\hat{\pi} = q_{\sigma}$, we have

$$V_{\mathsf{M}}^{\hat{\pi}} = V_{\mathcal{G}}^{\sigma}, \tag{36}$$

which completes the proof of Theorem 1.

$\square$

We also give proofs of Theorems 2 and 3, however, these proofs are highly similar to Theorem 1 so we only highlight key differences.

**Proof of Theorem 2.**

Let $\Pi_{\mathcal{M}}$ define the policy space for the multi-environment POMDP (ME-POMDP), and let $\Pi_{\hat{\mathsf{M}}}$ be the policy space for the agent in the AB-POMDP. A policy $\pi \in \Pi_{\mathcal{M}}$ is a mapping

$$\pi : (A \times Z)^* \to \Delta A, \tag{37}$$

while a policy $\sigma \in \Pi_{\hat{\mathsf{M}}}$ is a mapping

$$\sigma : (A \times (Z \cup \{\top\}))^* \to \Delta A. \tag{38}$$

We define the same mapping $f : \Pi_{\mathcal{M}} \to \Pi_{\hat{\mathsf{M}}}$ as in Theorem 1. That is,

$$f(\pi)(a_1, z_1, a_2, z_2, \ldots, a_n, z_n) = \begin{cases} \pi(a_2, z_2, \ldots, a_n, z_n) & n \geq 2, \wedge z_2 \neq \top \wedge \ldots \wedge z_n \neq \top, \\ \pi(\epsilon) & n = 1, \\ \delta_{\Diamond} & \text{otherwise.} \end{cases} \tag{39}$$

For a fixed pair of policies, $\pi$ and $f(\pi)$, we claim that $V_{\hat{\mathsf{M}}}^{f(\pi)} = V_{\mathcal{M}}^{\pi}$. The value of $f(\pi)$, for a belief $b \in \Delta(\{\bot\} \times [n])$, is

$$\sum_{i \in [n]} b_{(\bot, i)} V_{\hat{\mathsf{M}}_{\delta_{(\bot, i)}}}^{f(\pi)}, \tag{40}$$

and as the belief set is the set of distributions on a set of states,

$$\min_{b \in B} \sum_{i \in [n]} b_{(\perp,i)} V^{f(\pi)}_{\hat{\mathsf{M}}_{\delta_{(\perp,i)}}} = \min_{b \in \Delta(\{\perp\} \times [n])} \sum_{i \in [n]} b_{(\perp,i)} V^{f(\pi)}_{\hat{\mathsf{M}}_{\delta_{(\perp,i)}}} = \min_{i \in [n]} V^{f(\pi)}_{\hat{\mathsf{M}}_{\delta_{(\perp,i)}}} . \tag{41}$$

The second equality simply uses the fact that if we minimize a function $\sum_{i=1}^{n} c_i x_i$ for $(x_i)_{i=1}^{n}$ in the set of distributions on $[n]$, the minimum is the smallest element in $(c_i)_{i=1}^{n}$.

We can write the value for a fixed initial state as

$$V^{f(\pi)}_{\hat{\mathsf{M}}_{\delta_{(\perp,i)}}} = \mathbb{E} \left[ \sum_{t=2}^{H+1} \gamma^{t-1} \hat{R}(\hat{s}_t, a_t) \right], \tag{42}$$

where

$$\hat{s}_1 = (\perp, i), \tag{43}$$

$$\hat{s}_{t+1} \sim \hat{T}(\hat{s}_t, a_t) \qquad\qquad \forall t \geq 1, \tag{44}$$

$$a_{t+1} \sim f(\pi)(a_1, z_1, a_2, z_2, \ldots, a_t, z_t) \qquad\qquad \forall t \geq 1, \tag{45}$$

$$z_t \sim \hat{O}(\hat{s}_{t+1}, a_t) \qquad\qquad \forall t \geq 1. \tag{46}$$

We ignore the first timestep reward as it is $0$. Using the definition of the AB-POMDP $\hat{\mathsf{M}}$, we can evaluate $V^{f(\pi)}_{\hat{\mathsf{M}}_{\delta_{(\perp,i)}}}$. Let $\hat{s}_t$ denote the state in the AB-POMDP at time $t$. We have $\hat{s}_2 = (s_2, i, 1)$ where $s_2 \sim b_i$. For all $t \geq 3$, we then have $\hat{s}_t = (s_t, i, 2)$ where $s_{t+1} \sim T_i(s_t, a_t)$. By construction of the AB-POMDP, we have $a_1 = \Diamond$, and $z_1 \sim \hat{O}(\hat{s}_2, a_1) = \hat{O}((s_2, i, 1), \Diamond) = \delta_\top$, so again, we have $f(\pi)(a_1, z_1, a_2, z_2 \ldots, a_t, z_t) = \pi(a_2, z_2, \ldots, a_t, z_t)$, and $f(\pi)(\Diamond, \top) = \pi(\epsilon)$. For all $t \geq 2$ we have $z_t \sim \hat{O}(\hat{s}_{t+1}, a_t) = \hat{O}((s_{t+1}, i, 2), a_t) = O_i(s_{t+1}, a_t)$. Finally, we have $\hat{R}(\hat{s}_t, a_t) = \hat{R}((s_t, i, j), a_t) = R_i(s_t, a_t)/\gamma$ for all $t \geq 2$. Thus, we can write the value as

$$V^{f(\pi)}_{\hat{\mathsf{M}}_{\delta_{(\perp,i)}}} = \mathbb{E} \left[ \sum_{t=2}^{H+1} \gamma^{t-2} R_i(s_t, a_t) \right], \tag{47}$$

where

$$s_2 \sim b_i, \tag{48}$$

$$s_{t+1} \sim T_i(s_t, a_t) \qquad\qquad \forall t \geq 2, \tag{49}$$

$$a_2 \sim \pi(\epsilon) \tag{50}$$

$$a_{t+1} \sim \pi(a_2, z_2, \ldots, a_t, z_t) \qquad\qquad \forall t \geq 2, \tag{51}$$

$$z_t \sim O_i(s_{t+1}, a_t) \qquad\qquad \forall t \geq 2. \tag{52}$$

Finally, by the same state and timestep relabeling approach we used in Theorem 1, we obtain

$$V^{f(\pi)}_{\hat{\mathsf{M}}_{\delta_{(\perp,i)}}} = V^{\pi}_{\mathcal{M}_i}. \tag{53}$$

Thus, we have

$$\min_{b \in \Delta(\{\perp\} \times [n])} \sum_{i \in [n]} b_{(\perp,i)} V^{f(\pi)}_{\hat{\mathsf{M}}_{\delta_{(\perp,i)}}} = \min_{i \in [n]} V^{f(\pi)}_{\hat{\mathsf{M}}_{\delta_{(\perp,i)}}} = \min_{i \in [n]} V^{\pi}_{\mathcal{M}_i}. \tag{54}$$

That is, the mapping $f$ preserves the value of the policy between the ME-POMDP and the AB-POMDP, and so we have proven $V_{\hat{\mathsf{M}}} \geq V_{\mathcal{M}}$.

For the reverse direction, we can again use the same argument as in Theorem 1. Indeed, we can take any policy $\sigma \in \Pi_{\hat{\mathsf{M}}}$ and decompose it into policies $q_\sigma, r$ and $w$ where $q_\sigma$ is a policy in the ME-POMDP such that

$$q_\sigma(a_1, z_1, \ldots, a_t, z_t) = \sigma(\Diamond, \top, a_1, z_1, \ldots, a_t, z_t) \quad \forall (a_1, z_1, \ldots, a_t, z_t) \in (A \times Z)^*. \tag{55}$$

As the policies $f(q_\sigma)$ and $\sigma$ agree on all feasible histories in the AB-POMDP $\hat{\mathsf{M}}$, we deduce that

$$V^{\sigma}_{\hat{\mathsf{M}}} = V^{f(q_\sigma)}_{\hat{\mathsf{M}}} = V^{q_\sigma}_{\mathcal{M}}. \tag{56}$$

From this statement, we deduce that $V_{\hat{\mathsf{M}}} = V_{\mathcal{M}}$, and we have also implicitly proven, via the fact that $V_{\mathcal{M}}^{q_\sigma} = V_{\hat{\mathsf{M}}}^{\sigma}$, that the mapping from $\Pi_{\hat{\mathsf{M}}}$ to $\Pi_{\mathcal{M}}$ given in Theorem 2 preserves the value.

$\square$

**Proof of Theorem 3.**

The proof of this theorem follows the same framework as Theorems 1 and 2. That is,

1. We define a mapping $f : \Pi_{\mathsf{M}} \to \Pi_{\hat{\mathcal{M}}}$ as in Theorem 1.

2. We prove that $V_{\mathsf{M}}^{\pi} = V_{\hat{\mathcal{M}}}^{f(\pi)}$ for each $\pi \in \Pi_{\mathsf{M}}$.

3. We prove that, for each $\sigma \in \Pi_{\hat{\mathcal{M}}}$, the policy $q_\sigma$ defined by

   $$q_\sigma(a_1, z_1, \ldots, a_t, z_t) = \sigma(\Diamond, \top, a_1, z_1, \ldots, a_t, z_t) \quad \forall (a_1, z_1, \ldots, a_t, z_t) \in (A \times Z)^*,$$

   satisfies $V_{\mathsf{M}}^{q_\sigma} = V_{\hat{\mathcal{M}}}^{f(q_\sigma)} = V_{\hat{\mathcal{M}}}^{\sigma}$, and we conclude that $V_{\mathsf{M}} = V_{\hat{\mathcal{M}}}$, along with the fact that the mapping $\sigma \mapsto q_\sigma$ preserves value.

By assumption, the belief set $B$ is the set of distributions over $Q$, and so, for a policy $\pi \in \Pi_{\mathsf{M}}$,

$$\min_{b \in \Delta(Q)} V_{\mathsf{M}_b}^{\pi} = \min_{b \in \Delta(Q)} \sum_{q \in Q} b_q V_{\mathsf{M}_{\delta_q}}^{\pi} = \min_{q \in Q} V_{\mathsf{M}_{\delta_q}}^{\pi}. \tag{57}$$

Thus, the value of a policy $\pi$ in the AB-POMDP $\mathsf{M}$ is the worst-case value when we take a worst-case across the initial state.

However, this differentiation in the initial state is exactly how we define the partially observable MEMDP (PO-MEMDP) $\hat{\mathcal{M}}$. Indeed, for the same mapping $f : \Pi_{\mathsf{M}} \to \Pi_{\hat{\mathcal{M}}}$ as in the previous theorems, we can evaluate $V_{\hat{\mathcal{M}}_q}^{f(\pi)}$ as follows. Let $\hat{s}_t$ denote the sequence of states in the PO-MEMDP $\hat{\mathcal{M}}$. For environment $q$, we have $\hat{s}_2 = (q, 1)$, by definition of $\hat{T}$, and for all $t \geq 3$, we have $\hat{s}_t = (s_t, 2)$, where $s_{t+1} \sim T(s_t, a_t)$, again by definition of $\hat{T}$. We have $z_1 \sim \hat{O}(\hat{s}_2, \Diamond) = \hat{O}((q, 1), \Diamond) = \delta_\top$, and for all $t \geq 2$, we have $z_t \sim \hat{O}(\hat{s}_{t+1}, a_t) = \hat{O}((s_{t+1}, 2), a_t) = O(s_{t+1}, a_t)$. We additionally have, as in Theorem 1, $f(\pi)(a_1, z_1, a_2, z_2, \ldots, a_t, z_t) = \pi(a_2, z_2, \ldots, a_t, z_t)$, as $a_1 = \Diamond$ and $z_1 = \top$, and we also have $f(\pi)(a_1, z_1) = \pi(\epsilon)$. Finally, we have that $\hat{R}(\hat{s}_1, a_1) = 0$, and for all $t \geq 2$, $\hat{R}(\hat{s}_t, a_t) = \hat{R}((s_t, i_t), a_t) = R(s_t, a_t)/\gamma$. Thus, we have

$$V_{\hat{\mathcal{M}}_q}^{f(\pi)} = \mathbb{E}\left[ \sum_{t=2}^{H+1} \gamma^{t-2} R(s_t, a_t) \right], \tag{58}$$

where

$$s_2 = q \tag{59}$$
$$s_{t+1} \sim T(s_t, a_t) \qquad \forall t \geq 2 \tag{60}$$
$$z_t \sim O(s_{t+1}, a_t) \qquad \forall t \geq 2 \tag{61}$$
$$a_2 \sim \pi(\epsilon) \tag{62}$$
$$a_{t+1} \sim \pi(a_1, z_1, \ldots, a_t, z_t) \qquad \forall t \geq 2. \tag{63}$$

We can then again use the same relabelling steps as in the proofs of Theorems 1 and 2 to conclude

$$V_{\hat{\mathcal{M}}_q}^{f(\pi)} = V_{\mathsf{M}_{\delta_q}}^{\pi}. \tag{64}$$

We can then deduce the same value equivalence result, that is

$$V_{\mathsf{M}}^{\pi} = \min_{q \in Q} V_{\mathsf{M}_{\delta_q}}^{\pi} = \min_{q \in Q} V_{\hat{\mathcal{M}}_q}^{f(\pi)} = V_{\hat{\mathcal{M}}}^{f(\pi)}. \tag{65}$$

We then complete the same proof for step three as we used in the proofs of Theorems 1 and 2.

$\square$

**Proof of Theorem 4.**

As the PO-MEMDP $\mathcal{M}$ and multi-observation POMDP (MO-POMDP) $\hat{\mathcal{M}}$ share action and observation spaces, they have the same policy spaces.

Fix an arbitrary policy $\pi \in \Pi_{\mathcal{M}}$ and environment $i$. It is sufficient to show $V_{\mathcal{M}_i}^{\pi} = V_{\hat{\mathcal{M}}_i}^{\pi}$.

The proof then follows immediately by looking at the reward in the MO-POMDP. Denote the MO-POMDP state at time $t$ by

$$\hat{s}_t = (s_{1,t}, \ldots, s_{n,t}). \tag{66}$$

By definition of the MO-POMDP we have

$$V_{\hat{\mathcal{M}}_i}^{\pi} = \mathbb{E}\left[\sum_{t=1}^{H} \gamma^{t-1} \hat{R}_i(\hat{s}_t, a_t)\right] = \mathbb{E}\left[\sum_{t=1}^{H} \gamma^{t-1} R_i(s_{i,t}, a_t)\right] \tag{67}$$

where

$$s_{i,1} \sim b_i \tag{68}$$
$$s_{i,t+1} \sim T_i(s_{i,t}, a_t) \qquad\qquad \forall t \geq 1 \tag{69}$$
$$z_t \sim O(s_{i,t+1}, a_t) \qquad\qquad \forall t \geq 1 \tag{70}$$
$$a_1 \sim \pi(\epsilon) \tag{71}$$
$$a_{t+1} \sim \pi(a_1, z_1, \ldots, a_t, z_t) \qquad\qquad \forall t \geq 1. \tag{72}$$

However, up to labelling of the state random variable, this expectation is exactly $V_{\mathcal{M}_i}^{\pi}$.

$\square$

## A.2 Multiple Reward Functions are Necessary for Theorem 4

We next show that the presence of multiple reward functions is necessary for MO-POMDPs to simulate PO-MEMDPs and hence ME-POMDPs in the finite-horizon case. Previous reductions have shown that we can construct policies for one class, such as ME-POMDPs, by copying optimal policies from another class, such as PO-MEMDPs. We argue that no such reduction exists for MO-POMDPs when the reward does not vary with the environment. This argument appeals to the case where the environment has a single observation. With a single observation function, the multiple observation functions must be trivial, mapping all state-action pairs to one observation. Thus, as the reward function does not change with the environment, the MO-POMDP becomes a POMDP and has a history-dependent deterministic optimal policy. Meanwhile, PO-MEMDPs exist with a single observation and randomized optimal policies. Hence, there exist PO-MEMDPs such that no MO-POMDP produces a correct optimal policy. We formalize this argument in the following proposition.

**Proposition 1** *Consider the PO-MEMDP* $\mathcal{M} = (\{s\}, \{a_1, a_2\}, \{z\}, 2, \{T_i\}_{i \in [2]}, O, \{R_i\}_{i \in [2]}, \delta_s, 1, 1)$ *where*

$$R_1(s, a_1) = 1, \tag{73}$$
$$R_1(s, a_2) = -1, \tag{74}$$
$$R_2(s, a_1) = -1, \tag{75}$$
$$R_2(s, a_2) = 1, \tag{76}$$

*and* $T_i$ *and* $O$ *are defined appropriately. There does not exist a MO-POMDP* $\hat{\mathcal{M}} = (\hat{S}, \hat{A}, \hat{Z}, \hat{T}, \{\hat{O}_i\}_{i \in [n]}, \hat{R}, \hat{b}, \gamma, 1)$ *with an isomorphic observation and action space to* $\mathcal{M}$, *such that all optimal policies for* $\hat{\mathcal{M}}$ *are optimal for* $\mathcal{M}$.

**Proof:** The proof of this statement is mostly described in the preceding paragraph, but we elaborate it for completeness.

The PO-MEMDP $\mathcal{M}$ is a $2 \times 2$ matrix game, and has as a unique optimal policy

$$\pi^*(\epsilon) = \begin{cases} a_1 & \text{with probability } 1/2 \\ a_2 & \text{with probability } 1/2. \end{cases} \tag{77}$$

As the horizon of $\mathcal{M}$ is 1, the policy class comprises distributions on actions, and does not depend on the observation.

The policies in $\Pi_{\hat{\mathcal{M}}}$ are also a distribution over actions. However, as there is a single reward function, the reward of any policy is just the expected reward when we take the expectation over the action and the initial state. Thus, there exists an optimal policy for $\mathcal{M}$ that takes an action deterministically.

□

The above argument is similar to the remark in [4] that one can not use POMDPs to solve POSGs due to the different types of optimal policies they possess.

We remark that even if we added a finite number of extra steps to the MO-POMDP, as we do in Theorem 2 for the reduction from ME-POMDPs to AB-POMDPs, the observations at these timesteps would still need to be trivial to enable translation of MO-POMDP policies to the PO-MEMDP. Thus, even with these added timesteps, the policy would not depend on observations, and a policy would consist of a distribution of action sequences. There would then exist an optimal policy that deterministically follows an action sequence that attains the maximum expected reward.

We additionally remark that Proposition 1 can also be proven when we replace the PO-MEMDP $\mathcal{M}$ with a PO-MEMDP $\mathcal{M}$ that has multiple transition functions instead of multiple reward functions, but still, a single observation. Indeed, we simply add an extra step to $\mathcal{M}$, and define two extra states $s_1$ and $s_2$, such that the multiple transition functions change which action from $\{a_1, a_2\}$ leads to which state. We then define a reward of $+1$ for one of these states and a reward of $-1$ for the other, such that each action always leads to a reward of $+1$ in one environment and a reward of $-1$ in the other. These environments create the same problem as in Proposition 1 where the environment swaps the reward associated with each action, and so we require random optimal strategies.

### A.3  Proofs from Section 5.1

**Proof of Theorem 5**  Let $\hat{\pi}$ be the randomized policy we describe in Theorem 5, and let $v^*$ be the optimal value of (4). For any $b \in \Delta(Q)$ we have

$$V_{\mathsf{M}_b}^{\hat{\pi}} = \sum_{\alpha \in \Gamma} y(\alpha) V_{\mathsf{M}_b}^{\mathsf{Pol}(\alpha)} = \sum_{\alpha \in \Gamma} y(\alpha) \sum_{s \in S} \alpha(s) b(s) \geq \sum_{s \in S} v^* b(s) = v^*. \tag{78}$$

The first equality uses the definition of $\hat{\pi}$. The second equality uses the assumption on Pol. The third equality uses the feasibility of $y$ and the fact that $b$ only has support in $Q$. The final equality uses the fact that $b$ is a distribution. Thus we deduce $V_{\mathsf{M}}^{\hat{\pi}} \geq v^*$. However, by the duality of the LPs (3) and (4), $v^*$ is also the optimal value of (3), and so $v^*$ is an upper bound on the value of any policy. Hence, we conclude that $\hat{\pi}$ attains the optimal value of the AB-POMDP. □

### A.4  Constructing Behavioral Policies from Mixed Policies

Theorem 5 gives a recipe for constructing a mixed policy in an AB-POMDP which attains the value. In this section, we recall Kuhn's theorem [26] to give an explicit construction of an equivalent behavioral AB-POMDP policy.

Suppose a finite set $\mathcal{P} = \{\pi_1, \ldots, \pi_m\}$ of deterministic history-dependent policies is given, and let $\{p_i\}_{i=1}^m$ be a distribution over these policies that defines a mixed policy $\pi$.

We can construct a behavioral policy by randomizing over the deterministic policies in $\mathcal{P}$ that could have generated the current history. For a given finite-length history $h_t = (a_1, z_1, \ldots, a_t, z_t)$ in $(A \times Z)^*$, define $T(a_1, z_1, \ldots, a_t, z_t)$ with

$$T(a_1, z_1, \ldots, a_t, z_t) = \{i \in [m] \,|\, \forall k \in [t-1] : \pi_i(a_1, z_1, \ldots, a_k, z_k) = a_{k+1}\}. \tag{79}$$

$T(h_t)$ contains the indices of the deterministic policies in $\mathcal{P}$ that could possibly generate a history $h_t$. Define a behavioral policy as in [26] by

$$\pi_{\mathcal{P}, p}(h_t)(a) = \begin{cases} \frac{\sum_{i \in T(h_t) : \pi_i(h_t) = a} p_i}{\sum_{i \in T(h_t)} p_i} & T(h_t) \neq \emptyset \\ \Diamond & \text{otherwise,} \end{cases} \tag{80}$$

where $\Diamond$ is a fixed action in $A$.

The policies $\pi$ and $\pi_{\mathcal{P},p}$ have the same value for any POMDP. Indeed, the result in [26] implies that the distribution on finite-length histories is the same.

**Proposition 2 ([26])** *Let $\mathcal{P} = \{\pi_1, \ldots, \pi_m\}$ and $\{p_i\}_{i=1}^m$ define a mixed policy $\pi$, and let $\pi_{\mathcal{P},p}$ be the behavioral policy in* (80). *Then any finite length path $(s_1, a_1, z_1, \ldots, s_t, a_t, z_t)$ has the same probability under the mixed policy $\pi$ and the behavioral policy $\pi_{\mathcal{P},p}$.*

The proof of this statement follows by expanding the conditional probabilities that define the path. Thus, for any finite $l \leq H$ we have

$$\mathbb{E}_\pi \left[ \sum_{t=1}^l \gamma^{t-1} R(s_t, a_t) \right] = \mathbb{E}_{\pi_{\mathcal{P},p}} \left[ \sum_{t=1}^l \gamma^{t-1} R(s_t, a_t) \right]. \tag{81}$$

In the infinite-horizon case, we have

$$\lim_{l \to \infty} \mathbb{E}_\pi \left[ \sum_{t=1}^l \gamma^{t-1} R(s_t, a_t) \right] = \lim_{l \to \infty} \mathbb{E}_{\pi_{\mathcal{P},p}} \left[ \sum_{t=1}^l \gamma^{t-1} R(s_t, a_t) \right]$$

$$\implies \mathbb{E}_\pi \left[ \lim_{l \to \infty} \sum_{t=1}^l \gamma^{t-1} R(s_t, a_t) \right] = \mathbb{E}_{\pi_{\mathcal{P},p}} \left[ \lim_{l \to \infty} \sum_{t=1}^l \gamma^{t-1} R(s_t, a_t) \right]$$

$$\implies \mathbb{E}_\pi \left[ \sum_{t=1}^\infty \gamma^{t-1} R(s_t, a_t) \right] = \mathbb{E}_{\pi_{\mathcal{P},p}} \left[ \sum_{t=1}^\infty \gamma^{t-1} R(s_t, a_t) \right]. \tag{82}$$

We can swap limits and expectations here as the inner random variable is always bounded between $L/1-\gamma$ and $U/1-\gamma$, where $L$ is a lower bound on rewards and $U$ is an upper bound [34]. Thus, we conclude that the two policies have the same expected infinite-horizon reward.

## A.5 Constructing Mixed Policies from Approximate Value Functions

We must validate that the approximate value function and the associated $\alpha$-vectors satisfy the assumptions of Theorem 5, that for each $\alpha \in \Gamma$, there exists a policy $\pi$ such that

$$V_{\mathsf{M}_b}^\pi \geq \alpha \cdot b \quad \forall b \in \Delta(S). \tag{83}$$

Throughout this section, we will use

$$\mathbb{P}(s', z | s, a) = T(s, a)(s') O(s', a)(z) \tag{84}$$

as a shorthand for the probability of transitioning to a next state $s'$ and seeing an observation $z$, when the agent takes action $a$ at state $s$.

In a point-based backup, we start with an initial set of $\alpha$-vectors, and then we define each new $\alpha$-vector with an action $a$ and a mapping from each observation $z$ to some previously generated $\alpha$-vector $\alpha_z$. That is,

$$\alpha(s) = R(s, a) + \gamma \sum_{s' \in S, z \in Z} \mathbb{P}(s', z | s, a) \alpha_z(s'), \tag{85}$$

For each $\alpha$-vector, we can define two functions $\mathsf{Act} : \Gamma \to A$ and $\mathsf{Next} : \Gamma \times Z \to \Gamma$, that return the action and $\alpha$-vectors that defined it. For $\alpha$-vectors that are not the initial $\alpha$-vector, we define $\mathsf{Act}$ and $\mathsf{Next}$ such that $\mathsf{Act}(\alpha) = a$ and $\mathsf{Next}(\alpha, z) = \alpha_z$ in (85).

For the initial set of $\alpha$-vectors, we define an $\alpha$-vector $\alpha_{a,0}$ for each action $a$, that represents the policy that deterministically plays action $a$ at every time step, as in [41]. To compute the $\alpha$-vector $\alpha_{a,0}$, we start with the $\alpha$-vector that assigns the reward of the worst-case state for each action to each state, that is

$$\alpha_{a,0}(s) = \frac{\min_{s' \in S} R(s', a)}{1 - \gamma}. \tag{86}$$

We then improve each $\alpha_{a,0}$ by applying the following value iteration until the desired convergence.

$$\alpha_{a,0}(s) \leftarrow R(s, a) + \gamma \sum_{s'} T(s, a)(s') \alpha_{a,0}(s'). \tag{87}$$

By starting this iteration with the worst-case state underapproximation, HSVI ensures that $\alpha_{a,0}$ is a correct underapproximation regardless of the number of iterations of $\alpha_{a,0}$. We set

$$\mathsf{Act}(\alpha_{a,0}) = a, \quad \mathsf{Next}(\alpha_{a,0}, z) = \alpha_{a,0} \quad \forall z \in Z. \tag{88}$$

We note that with this $\alpha$-vector definition we have, for any initial state, that

$$\alpha_{a,0}(s) \leq \mathbb{E}_{a_t = \mathsf{Act}(\alpha_{a,0})} \left[ \sum_{t=1}^{\infty} \gamma^{t-1} R(s_t, a_t) | s_1 = s \right], \tag{89}$$

and hence this $\alpha$-vector lower bounds the value of the policy of always playing $\mathsf{Act}(\alpha_{a,0})$.

We can use these mappings to define a policy by tracking a current $\alpha$-vector as a state. Indeed, Algorithm 2 gives a recursive definition of a policy using the Act and Next functions. This method of extracting policies corresponds to the finite-state machine policy design in [17, 20].

---

**Algorithm 2** Extracting policies from $\alpha$-vectors.

    **procedure** EXECUTE($h = (a_1, z_1, \ldots, a_t, z_t), \alpha$)
        **if** $t = 0$ **then**
            **return** $\mathsf{Act}(\alpha)$
        **else**
            **return** EXECUTE($h = (a_2, z_2, \ldots, a_t, z_t), \mathsf{Next}(\alpha, z_1)$)
        **end if**
    **end procedure**

---

The policy we define in Algorithm 2 satisfies the requirements for Theorem 5. Indeed, we have the following proposition.

**Proposition 3** *Let* $\mathsf{Pol} : \Gamma \to \Pi$ *be the mapping given by Algorithm 2 that defines how to extract a policy from a set of $\alpha$-vectors. Then for all $\alpha \in \Gamma$, $s_0 \in S$, we have $\alpha(s_0) \leq V_{\mathsf{M}}^{\mathsf{Pol}(\alpha)}(s_0)$, and so $\alpha \cdot b \leq V_{\mathsf{M}_b}^{\mathsf{Pol}(\alpha)}$ for all $b \in \Delta(Q)$.*

This lemma specializes Proposition 9.7 in [22], which describes how to recover policies from approximate value functions in one-sided POSGs. The resulting proof is simpler than in [22], as the result we give only needs to hold for POMDPs.

**Proof of Proposition 3** We go by induction on the order in which we generate the $\alpha$-vectors. Let $\Gamma_k$ be the first $k \geq |A|$ generated $\alpha$-vectors.

For the base case, when $k = |A|$, $\Gamma_k$ contains the initial $\alpha$-vectors $\{\alpha_{a,0} | a \in A\}$. By the definition of $\mathsf{Pol}$, $\mathsf{Pol}(\alpha_{a,0})$ is simply the policy which always takes action $\mathsf{Act}(\alpha_{a,0}) = a$, and by (89) we have, for all $s_0$, that

$$\alpha_{a,0}(s_0) \leq V_{\mathsf{M}}^{\mathsf{Pol}(\alpha_{a,0})}(s_0). \tag{90}$$

Now, let $k$ be arbitrary, and suppose that, for all $\alpha \in \Gamma_k$, $s_0 \in S$, we have $V_{\mathsf{M}}^{\mathsf{Pol}(\alpha)}(s_0) \geq \alpha(s_0)$. Let $\alpha_{k+1}$ be the $\alpha$-vector generated next. When the agent plays $\mathsf{Pol}(\alpha_{k+1})$, we play $\mathsf{Act}(\alpha_{k+1})$, observe some $z$, and then play $\mathsf{Pol}(\mathsf{Next}(\alpha_{k+1}, z))$ from the next state. Thus, we can express the reward as follows

$$V_{\mathsf{M}}^{\mathsf{Pol}(\alpha_{k+1})}(s_0) = R(s_0, a) + \gamma \sum_{s' \in S, z \in Z} \mathbb{P}(s', z | s_0, a) V_{\mathsf{M}}^{\mathsf{Pol}(\mathsf{Next}(\alpha_{k+1}, z))}(s'). \tag{91}$$

Applying the inductive assumption, we then have $V_{\mathsf{M}}^{\mathsf{Pol}(\mathsf{Next}(\alpha_{k+1}, z))}(s') \geq \mathsf{Next}(\alpha_{k+1}, z)(s')$, as for all $z$, $\mathsf{Next}(\alpha_{k+1}, z) \in \Gamma_k$, and so we obtain

$$V_{\mathsf{M}}^{\mathsf{Pol}(\alpha_{k+1})}(s_0) \geq R(s_0, a) + \gamma \sum_{s' \in S, z \in Z} \mathbb{P}(s', z | s_0, a) \mathsf{Next}(\alpha_{k+1}, z)(s'), \tag{92}$$

and by definition of $\alpha_{k+1}$ through the backup we deduce that $V_{\mathsf{M}}^{\mathsf{Pol}(\alpha_{k+1})}(s_0) \geq \alpha_{k+1}(s_0)$. As $s_0$ was arbitrary, we now have that the hypothesis holds for the entire $\alpha_{k+1}$. Since $\Gamma_{k+1} = \Gamma_k \cup \{\alpha_{k+1}\}$

the hypothesis holds for all vectors in $\Gamma_{k+1}$. By the principle of mathematical induction, we then conclude that the lemma holds for all $\alpha$-vectors in $\Gamma$. $\square$

We remark that, as in [22], this procedure still works when pointwise dominated vectors are pruned. However, we must update the Next function in this case. Indeed, we define $\hat{\text{Next}}$ as an operator that returns some $\alpha$-vector that dominates the output of Next, *i.e.*,

$$\hat{\text{Next}}(\alpha, z)(s) \geq \text{Next}(\alpha, z)(s) \quad \forall s \in S, \alpha \in \Gamma, \tag{93}$$

and we define a corresponding policy in Algorithm 3.

---

**Algorithm 3** Extracting policies from $\alpha$-vectors with pruning.

**procedure** EXECUTE($h = (a_1, z_1, \ldots, a_t, z_t), \alpha$)
    **if** $t = 0$ **then**
        **return** $\text{Act}(\alpha)$
    **else**
        **return** EXECUTE($h = (a_2, z_2, \ldots, a_t, z_t), \hat{\text{Next}}(\alpha, z_1)$)
    **end if**
**end procedure**

---

A version of Proposition 3 still holds when we execute policies using Algorithm 3, but the analysis needs to be modified. The proof we give follows Proposition 9.7 in [22] again with simplifications due to the simpler setting of POMDPs.

**Proposition 4 ([22])** *Let* $\text{Pol} : \Gamma \to \Pi$ *be the mapping given by Algorithm 3 that defines how to extract a policy from a pruned set of $\alpha$-vectors. Then for all* $\alpha \in \Gamma$, $s_0 \in S$, *we have* $\alpha(s_0) \leq V_{\text{M}}^{\text{Pol}(\alpha)}(s_0)$.

**Proof of Proposition 4 ([22]):**

Let $U = \max_{s \in S, a \in A} R(s, a)$, $L = \min_{s \in S, a \in A} R(s, a)$, and $K = U - L$. $U$ is an upper bound on the reward, while $L$ is a lower bound on the reward.

Let $\text{Pol}_j(\alpha)$ define a policy such that, for the first $j$ actions, we follow Algorithm 3 and for the remaining timesteps, we use some fixed action $\Diamond \in A$. We will first show that the reward of the policy $\text{Pol}_j(\alpha)$ starting from $s_0$ is approximately lower-bounded by $\alpha(s_0)$, where the approximation error decreases geometrically in $j$. We then take a limit in $j$ to prove that $\alpha(s_0)$ lower-bounds the reward of $\text{Pol}(\alpha)$.

We first show by induction, that for all $\alpha \in \Gamma$, $s_0 \in S$, and $j \in \{0\} \cup \mathbb{N}$, we have

$$V_{\text{M}}^{\text{Pol}_j(\alpha)}(s_0) \geq \alpha(s_0) - \frac{\gamma^j}{1 - \gamma} K. \tag{94}$$

We first note that any $\alpha$-vector is bounded above by $U/(1 - \gamma)$. This fact obviously holds for the initial $\alpha$-vectors. For any subsequent $\alpha$-vectors, we have by induction that

$$\alpha(s) = R(s, a) + \gamma \sum_{s' \in S, z \in Z} \mathbb{P}(s', z | s, a) \alpha_z(s') \leq U + \gamma \frac{U}{1 - \gamma} = \frac{U}{1 - \gamma}, \tag{95}$$

where the induction is with respect to the order in which we generate $\alpha$-vectors.

For the main induction, that is (94), we have as our base case

$$V_{\text{M}}^{\text{Pol}_0(\alpha)}(s_0) = \mathbb{E}_{a_t = \Diamond} \left[ \sum_{t=1}^{\infty} \gamma^{t-1} R(s_t, a_t) \right] \geq \sum_{t=1}^{\infty} \gamma^{t-1} L = \sum_{t=1}^{\infty} \gamma^{t-1} (L - U) + \sum_{t=1}^{\infty} \gamma^{t-1} U$$

$$= \sum_{t=1}^{\infty} \gamma^{t-1} (L - U) + \frac{U}{1 - \gamma} \geq \left( \sum_{t=1}^{\infty} \gamma^{t-1} (L - U) \right) + \alpha(s_0) = \alpha(s_0) - \frac{1}{1 - \gamma} K. \tag{96}$$

The inequalities follow from the definitions of $L$ and $U$ along with the fact that the $\alpha$-vectors are bounded by $U/1-\gamma$.

Inducting on $j$, for an arbitrary $\alpha \in \Gamma$, we then have

$$V_{\mathsf{M}}^{\mathsf{Pol}_j(\alpha)}(s_0) = R(s_0, a) + \gamma \sum_{s' \in S, z \in Z} \mathbb{P}(s', z | s_0, a) V_{\mathsf{M}}^{\mathsf{Pol}_{j-1}(\hat{\mathsf{Next}}(\alpha, z))}(s')$$

$$\geq R(s_0, a) + \gamma \sum_{s' \in S, z \in Z} \mathbb{P}(s', z | s_0, a) \left( \hat{\mathsf{Next}}(\alpha, z)(s') - \frac{\gamma^{j-1}}{1-\gamma} K \right)$$

$$\geq R(s_0, a) + \gamma \sum_{s' \in S, z \in Z} \mathbb{P}(s', z | s_0, a) \left( \mathsf{Next}(\alpha, z)(s') - \frac{\gamma^{j-1}}{1-\gamma} K \right)$$

$$= \alpha(s_0) - \frac{\gamma^j}{1-\gamma} K. \quad (97)$$

The first equality is just expanding one step of the value. The first inequality uses the inductive assumption $V_{\mathsf{M}}^{\mathsf{Pol}_{j-1}(\alpha)}(s') \geq \alpha(s') - K\gamma^{j-1}/1-\gamma$. The second inequality uses the definition of $\hat{\mathsf{Next}}$. The final equality uses the definition of $\alpha$ through the backup.

We can then use a similar bound between $V_{\mathsf{M}}^{\mathsf{Pol}_j(\alpha)}$ and $V_{\mathsf{M}}^{\mathsf{Pol}(\alpha)}$. Indeed, as these policies share actions for the first $j$ timesteps, and rewards are bounded, we have

$$V_{\mathsf{M}}^{\mathsf{Pol}(\alpha)}(s_0) \geq V_{\mathsf{M}}^{\mathsf{Pol}_j(\alpha)}(s_0) - \frac{\gamma^j}{1-\gamma} K, \quad (98)$$

and combining the inequalities (97) and (98), we get

$$V_{\mathsf{M}}^{\mathsf{Pol}(\alpha)}(s_0) \geq \alpha(s_0) - \frac{2\gamma^j}{1-\gamma} K, \quad (99)$$

and we can take a limit in $j$ to conclude that we have the desired inequality. □

# B  Endangered Birds Preservation Model

Consider a population of an endangered bird species, as in [8]. The progression of the population is influenced by natural causes, such as feral cats hunting the endangered birds. We model the population of endangered birds under different actions to influence the progression of the population. At any time, we classify the population as being either low ($s_L$), middle ($s_M$), or high ($s_H$), although we assume we can only observe whether the population is high ($o_H$) or low ($o_L$). We consider two possible actions to influence the progression of the population level: control the feral cats (C), or do nothing (DN). The goal is to increase the population level to high and keep it there without wasting resources. For this purpose, we associate a reward to each state-action pair: $R(s_L, \mathrm{C}) = -5, R(s_L, \mathrm{DN}) = 0, R(s_M, \mathrm{C}) = 0, R(s_M, \mathrm{DN}) = 5, R(s_H, \mathrm{C}) = 5$, and $R(s_H, \mathrm{DN}) = 10$.

To define the probabilities in our model, we consider domain experts who define the transition and observation functions. For simplicity, we assume the experts all agree on the effect of doing nothing on the progression of the population of birds. See the table in Figure 5 for the different transition and observation probabilities the experts claim, where $p_{s_H} = 1 - (p_{s_L} + p_{s_M}), q_{s_H} = 1 - (q_{s_L} + q_{s_M}), w_{s_H} = 1 - (w_{s_L} + w_{s_M})$, and $O(s_M, a, o_L) = z_L, O(s_M, a, o_H) = 1 - z_L$ for $a \in \{\mathrm{C,DN}\}$.

Expert$_1$ believes that controlling the feral cats is not very effective when the bird population is low, but becomes more effective as the bird population increases. Additionally, Expert$_1$ believes a middle population level of the birds is just as likely to be interpreted as a high level and as a low level. Although Expert$_2$ agrees with Expert$_1$ on the observation probability, Expert$_2$ believes that controlling feral cats is most effective when the bird population is low, and becomes as effective as doing nothing as the bird population increases. Based on these two experts, this problem can naturally be modeled as a PO-MEMDP, since we only have multiple transition functions.

Another expert, Expert$_3$, agrees with Expert$_1$ on the effectiveness of controlling the feral cats but believes that a middle population level of birds is more likely to be interpreted as high than low. If we only have Expert$_1$ and Expert$_3$, this problem can naturally be modeled as a MO-POMDP, since we only have multiple observation functions. If instead, we have all three experts, this problem should be modeled as a ME-POMDP, as we have both multiple transition and multiple observation functions.

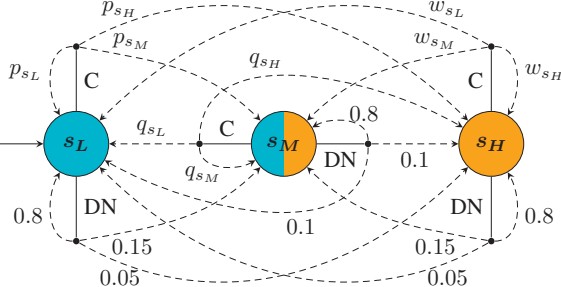

| unk. | Expert$_1$ | Expert$_2$ | Expert$_3$ |
|------|-----------|-----------|-----------|
| $p_{s_L}$ | 0.6 | 0.2 | 0.6 |
| $p_{s_M}$ | 0.35 | 0.6 | 0.35 |
| $q_{s_L}$ | 0.1 | 0.1 | 0.1 |
| $q_{s_M}$ | 0.5 | 0.75 | 0.5 |
| $w_{s_L}$ | 0 | 0.05 | 0 |
| $w_{s_M}$ | 0.1 | 0.15 | 0.1 |
| $z_L$ | 0.5 | 0.5 | 0.4 |

Figure 5: A visualisation of the Bird problem (left) with three experts (right). An action and corresponding transition distribution is represented by a solid line labeled by the action to a small node and dashed lines from the small node to the successor states labeled by the transition probabilities.

## C  Detailed Benchmark Descriptions

### C.1  Bird problem

We extend the endangered bird preservation example in Appendix B to arbitrary ME-POMDPs, PO-MEMDPs, and MO-POMDPs. In particular, we parameterize the number of states $|S| \geq 2$, actions $|A|$, and experts $n$. Each problem has a low and high population level state $s_L, s_H \in S$, and all other states represent population levels ordered between low and high. Regardless of the action, $s_L$ is always observed as a low population, $s_H$ is always observed as a high population, and all other states are observed as either high or low populations with distinct observation probabilities. Each action can be taken from each state and transitions to the same state and the two states with the closest population levels. In case of $s_L$, these closest population levels are the two population levels above $s_L$, and similar for $s_H$ the two population levels below $s_H$. For all other states, the two closest population levels are the levels one above and one below the current state. We randomly define $|A|$ probability distributions over $\min(|S|, 3)$ elements with each probability a multiple of $0.05$. We order these $|A|$ probability distributions inverse lexographically, meaning the probability distribution that assigns the most probability to the lowest state, *i.e.*, the state representing the lowest population level, is first in the ordering. The first probability distribution is therefore considered the least effective, whereas the last is considered the most effective.

Each expert defines an ordering over the actions for each state. This ordering represents which action each expert considers the most effective. We randomly generate the orderings, ensuring that each expert has a different effectiveness ordering for at least one state. The probability distribution of an action in an environment is hence given by the probability distribution corresponding to the effectiveness ranking the expert assigns to that action.

We also randomly generate, for each expert, $|S| - 2$ probability distributions over the two observations $o_L$ and $o_H$, again with each probability a factor of $0.05$. We order these probability distributions lexographically, so the first probability distribution gives the most probability to $o_L$. Each observation probability distribution is then linked to the in-between population level states with the same rank in the ordering.

The reward for each state-action pair is computed as the reward for the population level of the state minus the cost of the action. Each problem contains the do nothing action (DN) and $|A| - 1$ other actions. The DN action is free, all other actions add a reward of $-5$. The population level reward is $i \cdot 5$ where $i$ is the rank of the population level.

Depending on the model type we want to generate, we generate one or $n$ orderings over actions for each state and one or $n$ observation probability distributions for $|S| - 2$ states.

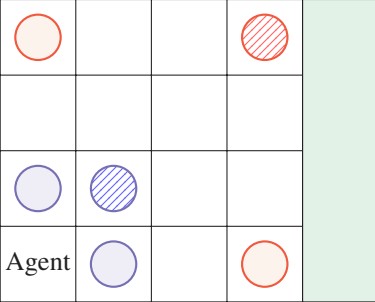

Figure 6: Visualization of relative fixed rock positions. Blue circles indicate the nearby positions, red circles the far away positions. The dashed circles are only considered with the three-rock version of the RockSample problem instances.

## C.2 RockSample

The second problem is based on RockSample [40]. The RockSample problem consists of an $m \times m$ grid with $t$ rocks at known positions. Each rock is either good or bad, but the agent does not know the state of the rocks. The agent starts in the bottom left corner, can move through the grid in the cardinal directions and can exit the grid at the rightmost positions. When in the same position as a rock, the agent can sample the rock. A good rock gives a reward of $10$, and a bad rock a reward of $-10$. After sampling a good rock, it becomes bad. To learn the state of a rock, the agent can check a rock. The probability that this check action gives the agent the correct state depends on the distance between the agent and the rock.

In the original RockSample problem, the agent starts with a belief that each rock has a 50% chance of being a good rock. We instead assume there are $g$ good rocks and $t - g$ bad rocks. We consider problem instances with randomly generated rock positions, as well as with relative fixed rock positions nearby or far away. For the RockSample instances with relative fixed positions, we consider either 2 or 3 rocks. The nearby rock positions are the three adjacent positions from the agent's starting positions, dropping the diagonal adjacent position for the two-rock version. The far-away rock positions are the three corners that the agent does not start in, dropping the rock in the top right corner for the two-rock version. The far-away rock positions change with the grid size, and are therefore relatively fixed. See Figure 6 for a visualization of the relative fixed rock positions.

We can define our version of the RockSample problem either as an AB-POMDP or a ME-POMDP.

To model our version of RockSample as an AB-POMDP, we only need to replace the initial belief of the original problem by the set of states where $g$ out of the $t$ rocks are good and the agent is in the bottom left corner. We can simplify this model slightly by removing the unreachable states in which more than $g$ rocks are good.

To model our version of RockSample as a ME-POMDP, we can simplify the state space to only keep track of the $g$ good rocks instead of all $t$ rocks. We then add $\binom{g}{t}$ environments, *i.e.*, one environment for each combination of good rocks and bad rocks. The state space can be used to keep track of the state of each good rock, whereas the environment maps the good rock state to an actual rock. Since the transition and observation functions depend on the state of each rock and therefore on the environment, we need to model this as a ME-POMDP. The initial state in each environment is the state where all good rocks are still good and the agent is in the bottom left corner.

# D Experimental Details

Table 4 shows the variation in convergence time and value for RockSample for environments with randomized rock positions. Table 4 shows similar trends to Table 2 whereby we see a significant increase in convergence time as we increase the number of environments. However, due to randomization of the rock positions in the instances, convergence time does not increase monotonically with the number of states, as can be seen by the convergence time decrease of instance $RS_{6,1,2}$ compared to instance $RS_{5,1,2}$.

Table 4: Lower bound value, convergence time, and gap for various RockSample problems.

| Model | $|S|$ | $n$ | $|A|$ | $|Z|$ | $V_{<tl}$ | Conv (s) | Gap |
|---|---|---|---|---|---|---|---|
| | **Properties** | | | | **AB- HSVI** | | |
| $RS_{3,1,2}$ | 19 | 2 | 7 | 3 | 15.55 | 41.68 | $< \epsilon$ |
| $RS_{3,1,3}$ | 19 | 3 | 8 | 3 | 14.17 | 774.99 | $< \epsilon$ |
| $RS_{3,1,4}$ | 19 | 4 | 9 | 3 | 14.37 | 2840.85 | $< \epsilon$ |
| $RS_{3,1,5}$ | 19 | 5 | 10 | 3 | 10.94 | - | 5.16 |
| $RS_{3,2,3}$ | 37 | 3 | 8 | 3 | 22.56 | 891.43 | $< \epsilon$ |
| $RS_{3,2,4}$ | 37 | 6 | 9 | 3 | 11.38 | - | 12.73 |
| $RS_{4,1,2}$ | 33 | 2 | 7 | 3 | 14.40 | 215.40 | $< \epsilon$ |
| $RS_{5,1,2}$ | 51 | 2 | 7 | 3 | 13.51 | 1252.50 | $< \epsilon$ |
| $RS_{6,1,2}$ | 73 | 2 | 7 | 3 | 14.09 | 1008.00 | $< \epsilon$ |
| $RS_{7,1,2}$ | 99 | 2 | 7 | 3 | 10.02 | - | 3.43 |

Table 5: Value and convergence time for RockSample problems modeled as AB-POMDPs and ME-POMDPs.

| Model | **AB-POMDP** | | **ME-POMDP** | |
|---|---|---|---|---|
| | $V_{<tl}$ | Conv (s) | $V_{<tl}$ | Conv (s) |
| $RS_{3,1,2}$ | 15.55 | **38.76** | 15.55 | 41.68 |
| $RS_{3,1,3}$ | 14.37 | 878.39 | 14.17 | **774.99** |
| $RS_{3,1,4}$ | 14.63 | 3064.78 | 14.37 | **2840.85** |
| $RS_{3,2,3}$ | 22.92 | 1269.64 | 22.56 | **891.43** |
| $RS_{4,1,2}$ | 14.40 | 267.28 | 14.40 | **215.40** |
| $RS_{5,1,2}$ | 13.51 | 1355.12 | 13.51 | **1252.50** |
| $RS_{6,1,2}$ | 14.09 | **949.19** | 14.09 | 1008.00 |

Table 6: The cost of robustness in RockSample. For each POMDP $M_i$, we report the value $V^{M_i}$, the time to solve all POMDPs, and the best and worst-case value under a misspecified environment, which we denote by $\overline{V}$ and $\underline{V}$, respectively. For the ME-POMDP we report the robust value $V$, the computation time, and the computation time factor.

| Model | **Individual POMDPs** | | | | | **Incorrect** | | **ME-POMDP** | | |
|---|---|---|---|---|---|---|---|---|---|---|
| | $V^{M_0}$ | $V^{M_1}$ | $V^{M_2}$ | $V^{M_3}$ | Time | $\overline{V}$ | $\underline{V}$ | $V$ | Time | Factor |
| $RS_{3,1,2}$ | 16.31 | 17.65 | | | 19.67 | -0.84 | -1.35 | 15.55 | 41.68 | 2.11 |
| $RS_{3,1,3}$ | 17.17 | 15.88 | 17.60 | | 43.82 | -0.41 | -0.88 | 14.17 | 774.99 | 17.68 |
| $RS_{3,1,4}$ | 17.17 | 17.65 | 17.60 | 18.07 | 43.65 | -0.45 | -1.35 | 14.37 | 2840.85 | 65.09 |
| $RS_{3,2,3}$ | 26.22 | 25.81 | 23.75 | | 163.61 | 9.07 | 5.70 | 22.56 | 891.43 | 5.49 |
| $RS_{4,1,2}$ | 16.76 | 15.09 | | | 100.22 | -0.39 | -1.29 | 14.40 | 215.40 | 2.15 |
| $RS_{5,1,2}$ | 15.13 | 14.78 | | | 258.66 | -1.16 | -1.51 | 13.51 | 1252.50 | 4.84 |
| $RS_{6,1,2}$ | 14.78 | 16.01 | | | 403.18 | -1.51 | -2.04 | 14.09 | 1008.00 | 2.50 |

Table 5 shows the variation in convergence time and value between modeling RockSample as an AB-POMDP or a ME-POMDP, as explained in Appendix C, for the Rocksample instances of Table 4 that converged within the time limit. Table 5 contains the data that defines Figure 3, which we discuss in Section 6. In particular, we note that in all but two models, the ME-POMDP formulation, which has multiple environments, converges faster the AB-POMDP formulation, which has a single environment. We hypothesize that this difference occurs because, in the ME-POMDP formulation, once an environment has a zero-probability, all beliefs in that environment can be ignored by a single check, whereas in the AB-POMDP formulation all zero-probability environment-state combinations must be checked separately.

Table 6 compares the value of solving a ME-POMDP and the time required, as compared to the naive baseline of solving the individual POMDPs, *i.e.*, the environments of the ME-POMDP. Note that Table 6 contains the data that defines Figure 4 and Table 3, which we discuss in Section 6. The robust value achieved by the ME-POMDP lies close to the optimal values of each individual POMDP, and far outperforms both the best and worst-case value achieved by assuming an incorrect environment as the true underlying environment, and playing the optimal policy accordingly. However, we also note that solving the ME-POMDP requires more time than the sum of solving all individual POMDPs. The factor with which the convergence time increases scales with the number of environments.

Table 7 compares the value and convergence time for the RockSample problem when we either place the rocks near or far from the agent's initial position. Table 7 extends Table 2 with results for the case when we have 2 good rocks, and these results corroborate the trends seen in Section 6.

We also depict the scaling of the solution times with the number of environments in Figure 7 for instances where the rocks are placed close to the agent's initial position.

Table 7: Lower bound value, time of convergence, and left-over gap between upper and lower bound of the RockSample problem for various problem sizes with rocks nearby or far away.

| Model | Properties | | | | Rocks nearby | | | Rocks far away | | | Factor |
|---|---|---|---|---|---|---|---|---|---|---|---|
| | $|S|$ | $n$ | $|A|$ | $|Z|$ | $V_{<tl}$ | Conv (s) | Gap | $V_{<tl}$ | Conv (s) | Gap | |
| $RS^c_{2,1,2}$ | 9 | 2 | 7 | 3 | 16.53 | 11.70 | $< \epsilon$ | 16.53 | 11.70 | $< \epsilon$ | 1 |
| $RS^c_{3,1,2}$ | 19 | 2 | 7 | 3 | 16.14 | 52.74 | $< \epsilon$ | 14.68 | 169.95 | $< \epsilon$ | 3.22 |
| $RS^c_{4,1,2}$ | 33 | 2 | 7 | 3 | 15.48 | 130.77 | $< \epsilon$ | 13.02 | 1588.97 | $< \epsilon$ | 12.15 |
| $RS^c_{5,1,2}$ | 51 | 2 | 7 | 3 | 15.40 | 331.37 | $< \epsilon$ | 11.03 | - | 1.46 | |
| $RS^c_{6,1,2}$ | 73 | 2 | 7 | 3 | 14.52 | 640.40 | $< \epsilon$ | | | | |
| $RS^c_{7,1,2}$ | 99 | 2 | 7 | 3 | 14.54 | 1280.66 | $< \epsilon$ | | | | |
| $RS^c_{2,1,3}$ | 9 | 3 | 8 | 3 | 15.90 | 115.11 | $< \epsilon$ | 15.90 | 115.11 | $< \epsilon$ | 1 |
| $RS^c_{3,1,3}$ | 19 | 3 | 8 | 3 | 15.41 | 269.10 | $< \epsilon$ | 14.34 | 1072.32 | $< \epsilon$ | 3.98 |
| $RS^c_{4,1,3}$ | 33 | 3 | 8 | 3 | 15.14 | 787.82 | $< \epsilon$ | 11.11 | - | 2.73 | |
| $RS^c_{5,1,3}$ | 51 | 3 | 8 | 3 | 14.80 | 1793.75 | $< \epsilon$ | 8.15 | - | 5.34 | |
| $RS^c_{6,1,3}$ | 73 | 3 | 8 | 3 | 14.31 | 2556.11 | $< \epsilon$ | | | | |
| $RS^c_{7,1,3}$ | 99 | 3 | 8 | 3 | 13.30 | - | 2.25 | | | | |
| $RS^c_{2,2,3}$ | 17 | 3 | 8 | 3 | 22.59 | 443.88 | $< \epsilon$ | 22.59 | 443.88 | $< \epsilon$ | 1 |
| $RS^c_{3,2,3}$ | 37 | 3 | 8 | 3 | 22.32 | 1105.12 | $< \epsilon$ | 19.50 | 3212.43 | $< \epsilon$ | 2.91 |
| $RS^c_{4,2,3}$ | 65 | 3 | 8 | 3 | 22.01 | 2389.88 | $< \epsilon$ | 13.15 | - | 6.64 | |
| $RS^c_{5,2,3}$ | 101 | 3 | 8 | 3 | 16.56 | - | 7.29 | | | | |

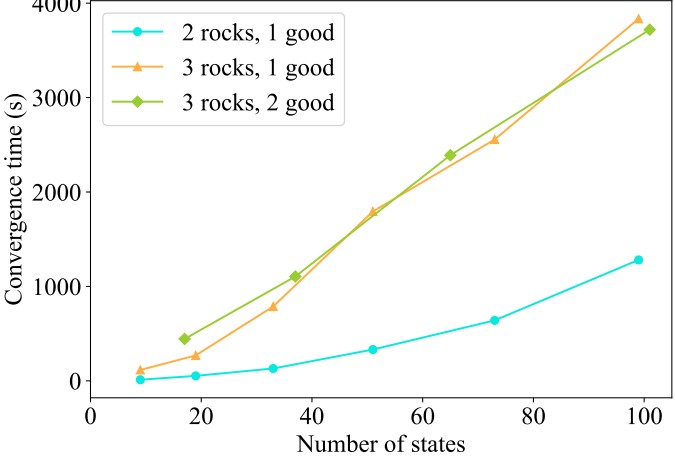

Figure 7: Variation of convergence time with the number of states in the RockSample instances with fixed rock positions nearby.

