# OpenReview forum: "Multi-Environment POMDPs: Discrete Model Uncertainty Under Partial Observability"
_NeurIPS.cc/2025/Conference — NeurIPS 2025 poster_

### Official Review · Reviewer_xCNX · 2025-06-30

**Clarity:** 4
**Significance:** 3
**Originality:** 4
**Rating:** 5
**Confidence:** 3

**Summary:**

This paper addresses finding a robust policy when there are a finite different models for a POMDP, where each model can have different transition, observation, and reward models. To solve this, they are optimizing for the policy that maximizes the expected value of the worst-case model. They show that solving this problem is equivalent to playing a partially observable adversarial game where the fully observable agent’s policy is to minimize the performance of the partially observable agent by selecting the model. They also provide a process of converting the set of models into a Multi-Environment POMDP, where the set of models is encoded into the problem and belief state. As such, the adversarial agent is selecting a belief state. They introduce an algorithm, Adversarial-Belief HSVI, for solving for the robust policy and evaluate this algorithm on two different domains.

**Questions:**

1.	Do you have a proof or empirical evaluation for showing that AB-HSVI converges over time?
2.	Have you evaluated any other problems or done additional evaluation on the computation cost of changing the individual factors?

**Ethical Concerns:**

["NO or VERY MINOR ethics concerns only"]

**Final Justification:**

The authors have addressed all questions and concerns raised by me and the other reviewers. To the best of my knowledge, the presented theory is sound. While additional theoretical analysis of their algorithm would further strengthen the work, the authors acknowledge the main weakness of their approach (computational cost) and have addressed it with sufficient empirical evaluation. Their empirical results are solid, and the additional experiments being added will provide valuable insight into their approach. Overall, this paper makes a meaningful contribution to the literature and would be a valuable addition to the conference.

**Limitations:**

The main limitations are discussed; however, having a limitation section would be beneficial.

**Paper Formatting Concerns:**

Formatting is good.

**Quality:**

4

**Strengths And Weaknesses:**

**Strengths:**

* The main contribution of this paper is the framework and theoretical results that are clear and sound, to the best of my knowledge.
* The paper clearly distinguishes itself from the existing work on robust policies, providing a convincing argument on its novelty.
* The methodology of the empirical evaluation is clear and reproducible.

**Weaknesses:**

* The paper currently does not have a proof or empirical evaluation showing that the AB-HSVI algorithm converges. It is clear that it does, just that explicitly including it would strengthen the paper.
* While the empirical evaluation effectively shows the behavior of the AB-HSVI algorithm and solving for the robust policy using the ME-POMDP, it is hard to say when solving for the robust policy is feasible.
* It would have been nice if a more substantial ablation study had been performed to evaluate the approximate computational complexity of the individual variables (number of experts, amount of disagreement between the models, etc.)

**Other Comments:**

* One assumption that I think needs to be mentioned is that the reward function needs to use the same units between models. Otherwise, comparing the expected value between the models is not a meaningful metric. For example, an expert could increase the survival cost in Rocksample such that their model dominates the set of models, causing the robust policy to have the agent exit as fast as possible.
* The performance and the magnitude of the differences between the models would impact the quality of the robust policy. For example, if the POMDP encodes a multi-objective problem where experts disagree on the objective, the robust policy is unlikely to achieve either objective.
* Table 4’s comparison of the robust policy compared to the optimal policy for each individual model is interesting but causes more confusion than benefits for the main paper. Since the robust policy solves a different problem, the only conclusion that can be drawn is that solving the optimal policy for an individual model will outperform the robust policy. Rather, it creates more confusion, suggesting that the robust policy is somehow one of these optimal policies.

---

> ### Author Rebuttal · Authors · 2025-07-31
>
> We would like to thank the reviewer for their helpful comments, suggestions, and questions.
> Based on those, we plan to make the following changes to our paper:
> - Include a more detailed discussion on the effects of assumptions made on the reward function.
> - Clarify the rationale behind Table 4 in the experimental evaluation.
> - Include new experimental results to provide more empirical guidance on expected runtimes in terms of model size.
> - Incorporate all other suggestions and directions for future work.
>
> We first address the comments.
>
> > C1. *One assumption that I think needs to be mentioned is that the reward function needs to use the same units between models.*
>
> We acknowledge that scaling in the reward functions of different experts/models is an interesting problem.
> For example, one could design multi-environment POMDP solvers that extract the qualitative behavior from different environments without being affected by the rewards.
> While such scaling problems may also be seen as a modeling problem, we will make sure that we add a discussion of this assumption to the paper.
>
>
> > C2. *The performance and the magnitude of the differences between the models would impact the quality of the robust policy. For example, if the POMDP encodes a multi-objective problem where experts disagree on the objective, the robust policy is unlikely to achieve either objective.*
>
> We agree that the quality of the robust policy depends on the differences between the environments.
> If environments differ a lot, the robust policy will be more conservative than if environments are more similar.
>
> Although our approach works for environments with completely contradicting reward functions, our goal is always to find a robust policy with respect to these reward functions, which does not require satisfying different objectives in different environments.
> Such multi-objective problems are nonetheless an interesting and relevant direction for future work.
>
>
>
>
>
> > C3. *Table 4’s comparison of the robust policy compared to the optimal policy for each individual model creates confusion, suggesting that the robust policy is somehow one of these optimal policies.*
>
>
> We understand the confusion highlighted by the reviewer and thank them for bringing this issue to our attention.
> We believe the comparison in Table 4 is important to give insight into the computational cost of being robust versus the potential loss of value of naive approaches, such as using a policy that is optimal for one of the environments without taking the other environments into account.
> We will clarify the rationale behind Table 4 and the values compared in the table.
>
>
>
> Next, we answer the questions.
>
> > Q1. *Do you have a proof or empirical evaluation for showing that AB-HSVI converges over time?*
>
>
>
> **In summary:** We believe that there are promising directions for proving convergence using similar methods as for partially observable stochastic games.
>
> Standard HSVI convergence proofs fail in our setting because they rely on a constant initial belief, whereas we change the initial belief throughout AB-HSVI. We believe that it will be sufficient to prove a form of a uniform convergence result for the value-function upper and lower bounds to show that we have convergence at all initial belief points.
>
> To prove the convergence of HSVI in a POSG setting, the authors of [1] augmented the standard HSVI approximation algorithm.
> The first augmentation is a new upper bound with Lipschitz continuity.
> The second augmentation is a modified convergence condition for the forward belief search.
> These modifications facilitate a proof that, informally, convergence of HSVI at a point implies convergence in a neighborhood of that point.
>
> We believe that applying these modifications to AB-HSVI provides a promising path to show uniform convergence of the value function in AB-HSVI.
>
>
> > Q2. *Have you evaluated any other problems or done additional evaluation on the computation cost of changing the individual factors?*
>
>
> *Note: This is a similar answer to the one we give to Reviewer 3 (sP8H) Q1 and Q2, which we repeat here for readability.*
>
> Tables 1 and 2 provide empirical guidance on the influence of increased state, action, and environment spaces, but we also acknowledge that this guidance is limited due to the strong influence of the different environments.
> To improve the empirical guidance, we ran additional experiments on RockSample instances where the relative position of the rocks is fixed either near or far from the agent's starting position.
> We conducted multiple experiments with these fixed rock positions, varying the number of rocks, the number of good rocks, and the grid size.
> With these experiments, we will improve the empirical guidance for:
> 1. The influence of only increasing the state space while keeping similar environments.
> 2. The influence of different environments with the same state, action, and number of environments.
>
> For 1., we see that the increase in computation time still depends on the type of consistent environments.
> For example, when we have only two rocks that are placed adjacent to the agent's starting position, we see a runtime increase factor between $2$ and $3$ per grid size increase.
> On the other hand, when we have three rocks that are placed in the corners of the grid, we see a runtime increase factor of about $10$.
>
> For 2., we observe that two variants of a problem instance differing only in their environments exhibit a running time variation of a factor between 3 and 10. We expect this gap to widen further with an increase in the state space.
> We will include these additional experiments and discussion in the final version of the paper.
>
> #### References
>
> [1] Karel Horák, Branislav Bosanský, Vojtech Kovarík, Christopher Kiekintveld: Solving zero-sum one-sided partially observable stochastic games. Artif. Intell. 316: 103838 (2023)

---

> > ### Comment · Reviewer_xCNX · 2025-08-01
> >
> > Thank you for your clarifications and for agreeing to address my concerns in the revision.
> >
> > I would emphasize that my concerns with the reward function are more than a scaling problem. An assumption that the reward functions share a structure or have been calibrated on the same utility scale is needed. Otherwise, it becomes unclear how applicable the robust policy would be in practice.
> >
> > After reading all the reviews and rebuttals, while theoretical guarantees would improve the paper, it remains a valuable contribution. I will therefore maintain my positive recommendation for acceptance.

---

> > > ### Author Response · Authors · 2025-08-04
> > >
> > > We thank the reviewer for their reply and agree with their concerns about the reward function.
> > > We will ensure that we emphasize the requirement of a common structure on the reward functions to achieve an applicable robust policy.
> > > We will add this discussion to the paper in a dedicated paragraph.

---

### Official Review · Reviewer_sP8H · 2025-07-02

**Clarity:** 2
**Significance:** 3
**Originality:** 3
**Rating:** 4
**Confidence:** 1

**Summary:**

This paper addresses robust policy planning in ME-POMDPs, where multiple POMDPs defined over the same state, action, and observation spaces differ in their dynamics and reward functions. The authors extend the theoretical framework by introducing AB-POMDPs, which generalize ME-POMDPs by allowing the initial belief to be adversarially selected from a set. The main theoretical contribution is showing that any ME-POMDP can be efficiently reduced to an AB-POMDP, and both can be formulated as special cases of one-sided POSGs. The paper develops both exact and approximate (point-based) algorithms for robust policy computation, notably extending HSVI with linear programming to handle adversarial initial beliefs.

**Questions:**

1. The experimental tables show that convergence time can grow rapidly with the number of environments and problem size, sometimes resulting in incomplete runs. Can the authors provide theoretical or empirical guidance on the expected runtime as a function of the number of environments, states, and actions?
2. How well does the approach scale as the state, action, or environment space increases?
3. How do the proposed algorithms perform empirically compared to robust POMDP methods that use continuous or convex uncertainty sets, or alternative robust optimization frameworks?
4. On line 19, the equation "$H+1=H$" requires clarification. On line 99, the superscript $\star$ is not defined.

**Ethical Concerns:**

["NO or VERY MINOR ethics concerns only"]

**Final Justification:**

Based on the experimental results supplemented by the authors, I believe this paper has the potential to be accepted. However, since my research focuses more on theoretical results, my confidence level is 1, so as not to overly influence the final decision.

**Limitations:**

Yes

**Quality:**

3

**Strengths And Weaknesses:**

**Strengths**: This paper demonstrates notable theoretical depth and clarity by providing a rigorous formalization of ME-POMDPs and AB-POMDPs, including precise definitions, clear reductions, and systematic connections to POSGs. The foundational theoretical results are well-motivated, and the approach of reducing robust planning problems to standard constructs such as POSGs and restricted POMDP variants enables both conceptual clarity and algorithmic leverage.

**Weaknesses**
1. The paper does not provide a theoretical analysis of time or space complexity with respect to key parameters (e.g., state/action space size, number of environments, planning horizon).
2. Results are reported on randomly generated environments without variance, error bars, or confidence intervals, limiting the interpretation of empirical robustness.
3. There is no in-depth discussion of how randomness in benchmark generation may affect results, nor is there a systematic exploration of whether variations between random seeds could lead to significantly different conclusions.

---

> ### Author Rebuttal · Authors · 2025-07-31
>
> We would like to thank the reviewer for their helpful comments, suggestions, and questions.
> Based on those, we plan to make the following changes to our paper:
> - Clarify the use of randomization to find non-trivial benchmarks and include additional experimental results to further highlight the effects of specific model instances on runtime.
> - Include new experimental results to provide more empirical guidance on expected runtimes in terms of model size.
> - Incorporate all other suggestions and directions for future work.
>
> We first address the weaknesses that do not have a corresponding question.
>
> > W2/W3. *Results are reported on randomly generated environments without variance, error bars, or confidence intervals, limiting the interpretation of empirical robustness.
> There is no in-depth discussion of how randomness in benchmark generation may affect results.*
>
>
> *Note: This is the same answer as we give to Reviewer 1 (aQSA) Q3, which we repeat here for readability.*
>
> **In summary:**
> No benchmarks exist for ME-POMDPs, and a straightforward adaptation of existing POMDP benchmarks is not trivial.
> Nevertheless, we provide additional experiments on variations of the RockSample problem.
>
> First, we would like to note that there are no existing benchmarks for ME-POMDPs.
> Removing the initial belief from a POMDP benchmark model often does not result in an interesting problem, as the worst-case environment can be trivial.
> For example, in the original RockSample problem, the initial distribution defines how many rocks are good and bad.
> The environment where all rocks are bad is trivially the worst-case environment.
>
> To create challenging, non-trivial problems for evaluating AB-HSVI, we employed randomization.
> We only use this approach to substitute for the hand-design of models, and thus our experiments are comparable to standard POMDP experiments, where one also evaluates a single instance of each problem (see, for instance, [1], where RockSample was introduced and also used in multiple separate configurations).
>
> Creating challenging problem instances is difficult.
> For example, for the Bird problems with $3$ states, $3$ actions, and $3$ experts, generating models with random seeds $0$ to $99$, we only got $35$ out of $100$ non-trivial PO-MEMDPs, $34$ non-trivial ME-POMDPs, and only $16$ non-trivial MO-POMDPS.
> For this classification, we considered a problem instance trivial if it could be solved within $30$ seconds.
>
> Even though we can generate multiple non-trivial variants of a problem instance, the values and running times are highly dependent on the environments, and these variants are therefore better considered as distinct problems rather than variants that we should compare.
> To demonstrate this, we ran some additional experiments on RockSample instances where the only change was the position of the rocks.
> The number of states, actions, and environments is fixed across the instances.
> The running time between two such variants of a problem instance varied by factors between $3$ and $10$, and we expect this gap to further increase with a larger state space.
>
> We will clarify the use of randomization in our paper and include these experiments to further highlight the effects of the specific model on runtime.
>
>
> Next, we answer the questions.
>
> > Q1/Q2. *Can the authors provide theoretical or empirical guidance on the expected runtime as a function of the number of environments, states, and actions?
> How well does the approach scale as the state, action, or environment space increases?*
>
> We first note that, from a theoretical perspective, we can guarantee little, as ME-POMDPs already contain POMDPs as a special case, and finding optimal policies that maximize the expected discounted reward is undecidable for POMDPs [2].
> For finite-horizon reward maximization, the problem is at least PSPACE-hard [3], but the exact theoretical complexity of solving ME-POMDPs is still open.
>
> Although Tables 1 and 2 are intended to give empirical guidance on the influence of increased state, action, and environment spaces, we also acknowledge that this guidance is limited due to the strong influence of the different environments.
> To improve the empirical guidance, we ran additional experiments on RockSample instances where the relative position of the rocks is fixed either near or far from the agent's starting position.
> We conducted multiple experiments with these fixed rock positions, varying the number of rocks, the number of good rocks, and the grid size.
> With these experiments, we will improve the empirical guidance for:
> 1. The influence of only increasing the state space while keeping similar environments.
> 2. The influence of different environments with the same state, action, and number of environments.
>
> For 1., we see that the increase in computation time still depends on the type of consistent environments.
> For example, when we have only two rocks that are placed adjacent to the agent's starting position, we see a runtime increase factor between $2$ and $3$ per grid size increase.
> On the other hand, when we have three rocks that are placed in the corners of the grid, we see a runtime increase factor of about $10$.
>
> For 2. (as also stated in our response to W2/3), we observe that two variants of a problem instance differing only in their environments exhibit a running time variation of a factor between 3 and 10. We expect this gap to widen further with an increase in the state space.
> We will include these additional experiments and discussion in the final version of the paper.
>
> > Q3. *How do the proposed algorithms perform empirically compared to robust POMDP methods that use continuous or convex uncertainty sets, or alternative robust optimization frameworks?*
>
> The only available robust POMDP solvers assume $(s,a)$-rectangularity, which is an independence assumption between the uncertainty of different state-action pairs.
> This is a property that the general multi-environment problems that we consider do not have.
> Consequently, none of the problems we evaluate on are $(s,a)$-rectangular.
> Making a non-rectangular model $(s,a)$-rectangular would give the adversary significantly more power, as it can choose a separate worst-case distribution for each state-action pair.
> The computed optimal agent policy will then also be significantly more conservative than necessary for the multi-environment problem.
> This makes a comparison with robust POMDP solvers ineffective.
>
> We further remark that assuming $(s,a)$-rectangularity does not make the problem necessarily easier to solve.
> In [4], robust optimization is employed to solve robust POMDPs, but due to the non-convexity of the underlying optimization problem, iterative approximations are required.
>
> > Q4. *On line 19, the equation $H + 1 = H$ requires clarification. On line 99, the superscript \* is not defined.*
>
> - Line 93 indicates that $H + 1 = \infty$, as in this case, $H = \infty$.
> - For a finite set $X$, we denote by $X^\*$ the set of finite sequences of elements from $X$.
> That is, $\*$ indicates the Kleene-star operator.
>
> We will add these corrections and clarifications to the paper.
>
>
> #### References
>
> [1] Trey Smith, Reid G. Simmons: Heuristic Search Value Iteration for POMDPs. UAI 2004
>
> [2] Omid Madani, Steve Hanks, Anne Condon:
> On the Undecidability of Probabilistic Planning and Infinite-Horizon Partially Observable Markov Decision Problems. AAAI/IAAI 1999
>
> [3] Christos H. Papadimitriou, John N. Tsitsiklis: The Complexity of Markov Decision Processes. Math. Oper. Res. 12(3): 441-450 (1987)
>
> [4] Murat Cubuktepe, Nils Jansen, Sebastian Junges, Ahmadreza Marandi, Marnix Suilen, Ufuk Topcu:
> Robust Finite-State Controllers for Uncertain POMDPs. AAAI 2021

---

> ### Comment · Reviewer_sP8H · 2025-08-04
>
> Thanks for the rebuttal. However, the author did not respond to the Weakness 1.

---

> > ### Author Response · Authors · 2025-08-04
> >
> > We hope to have responded to this weakness in our reply to Questions 1 and 2, where we mention both theoretical complexity bounds, as well as empirical results indicating AB-HSVI's scalability in terms of states, actions, and number of environments. We are happy to provide further clarification if needed.

---

> > > ### Comment · Reviewer_sP8H · 2025-08-05
> > >
> > > I apologize for missing it. The further analysis and results in the rebuttal can improve the paper. I have no more questions.

---

### Official Review · Reviewer_g1P2 · 2025-07-02

**Clarity:** 2
**Significance:** 2
**Originality:** 3
**Rating:** 5
**Confidence:** 2

**Summary:**

This paper is about multi-environment POMDPs (ME-POMDPs), a version of POMDPs, in which the transition model, observation model or reward function is uncertain, in that there are a finite number of possible models. It is concerned with the setting without prior knowledge about which environment is active and tries to find policies that maximize the value in the worst case. This is achieved using an algorithm that transforms ME-POMDPs into so-called adversarial-belief POMDPs (AB-POMDPs) and solves the resulting AB-POMDPs using linear programming. The algorithm is evaluated on two benchmarks and compared to a baseline that solves the individual POMDP models without taking robustness into account.

**Questions:**

- Can't we write any ME-POMDP as a single POMDP, which includes an unobserved discrete variable indicating the true model, which does not change during an episode? If that is the case: What makes the current approach more efficient than the naive approach of solving that POMDP directly? And why was the naive approach not considered as a baseline?
- I understood the solution technique is to turn an ME-POMDP into an AB-POMDP and then solve that using AB-HSVI. If that is the case, then how do I interpret the comparison in Table 3 between ME-POMDPs and AB-POMDPs, both solved with AB-HSVI?
- In a POSG, why does the fully observing player's policy take the history of states, both players actions and observations as the input?
- In response to checklist item 7 (statistical significance), the authors state "The algorithm is deterministic. It does not use any randomization during runtime, and hence will produce the same result upon running it twice. Note, we do use randomization to generate the models that we run our algorithm on, but then the models are fixed." This argument is not convincing to me. Wouldn't we still want to know if the algorithm performs well on some distribution of models rather than on a single randomly generated model?

**Ethical Concerns:**

["NO or VERY MINOR ethics concerns only"]

**Final Justification:**

I appreciate the detailed responses by the authors to all of my questions and the outline for how the authors plan to improve the papers readability. The authors' explanation that there are no existing benchmarks for the multi-model POMDP setting was convincing to me and I think this paper will be a valuable contribution to the literature, which is why I have increased my score by one point. I would like to point out that I was not able to verify any of the proofs, and I have therefore kept my rather low confidence rating.

**Limitations:**

The paper does not discuss any limitations or potential societal impacts. In my opinion, a discussion of societal impacts is not necessary for this kind of basic theoretical research, but a discussion of limitations would be appreciated to put the approach into context.

**Paper Formatting Concerns:**

I have no concerns about formatting.

**Quality:**

3

**Strengths And Weaknesses:**

I appreciate that the authors have tried to present very technically dense material in a readable fashion. I found the introduction well-written and it seems to cover relevant related work. The section on POMDPs and alpha vectors provides useful background for readers not familiar with these concepts. However, after the introduction, the paper often feels less clear and I was at times confused about some points that seem quite central to the paper (see questions below), concerning both the results and the method. Furthermore, the paper lacks visualizations, intuitive explanations, or defnitions of some of the notation. For these reason, the rest of my review consists mostly of suggestions for how the paper could be made clearer and more accessible and of clarification questions.

1. The paper contains no figures at all. This is in some sense understandable for a theory paper, but I think the authors miss a few opportunities to present their work in a more accessible fashion. For example, the results would be more easily digestible if they were shown using a few clear figures instead of a whole page of tables. Also, a graphic illustrating the relationship between ME-POMDPs, PO-MEMDPs, MO-POMDPs and AB-POMDPs would be highly appreciated, as it was easy to lose track of how the concepts fit together.
2. I did not check the proofs in detail, as that is outside my area of expertise, but it seems to me that the theorems in Section 4 and 5 could benefit from some more intuitive descriptions. Before or after going into mathematical jargon, it would be nice to provide some intuition about what is actually being stated in natural language.
3. Symbols or equations are sometimes used before being properly defined or introduced. I will give a few examples, but this is not an exhaustive list. In Theorem 1, some of the symbols are not explained (e.g. ⊥, ⊤). On page 7, $y$ is used before being defined. The linear programs (3) and (4) are already referenced in Section 5.1, but are defined only Section 5.2, which in turn refers to "the LPs of Section 5.1".
4. The discussion of the results is very sparse and not easy to follow. See also my questions below.
    a. Section "Q1. Scalability." offers no discussion of the three different times of models (PO-MEMDP, MO-POMDP, and ME-POMDP).
    b. Statements like "the randomly generated environments have a great effect on the difficulty of the problems" would benefit from concrete pointers to numbers particular tables.

Minor points:
- The notation (A x Z)* for the history of states and observations makes sense but could be made more explicit that that is what is means, because in other contexts the authors use the star to indicate an optimal policy.
- Ordering the references by occurence would make for easier reading

---

> ### Author Rebuttal · Authors · 2025-07-31
>
> We would like to thank the reviewer for their helpful comments, suggestions, and questions.
> Based on those, we plan to make the following changes to our paper:
> - Clarify the results presented in Table 3.
> - Clarify the use of randomization to find non-trivial benchmarks and include additional experimental results to further highlight the effects of specific models on runtime.
> - Incorporate all other suggestions and directions for future work, paying particular attention to accessibility and readability. We will also highlight the limitations.
>
> We first clarify the meaning of the indicated symbols:
> - $\bot$ and $\top$ are fresh symbols used throughout model definitions to denote auxiliary states/observations.
> - $y$ is used to indicate a solution to the dual linear program.
>
> We will clarify these symbols in the final version of the paper.
>
> Next, we answer the questions.
>
> > Q1 *Can't we write any ME-POMDP as a single POMDP with an unobserved and discrete variable indicating the true environment? If so, what makes the current approach more efficient than solving that POMDP directly? And why was this naive approach not considered as a baseline?*
>
> Solving a ME-POMDP via a larger POMDP does not lead to robust policies.
> When we write a ME-POMDP $M$ as a larger POMDP $M'$, we must define an initial distribution (initial belief) in this larger POMDP $M'$.
> By defining this initial distribution, we define an initial distribution over the environments, and a policy found for this initial distribution will not be robust to other initial distributions.
>
> A simple way to show that an ME-POMDP is not equivalent to a larger POMDP is by noting that deterministic policies are optimal for finite-horizon POMDPs, but **not** for finite-horizon ME-POMDPs.
> Indeed, we can encode zero-sum matrix games via ME-POMDPs.
> We provide an example of this encoding in Proposition 1 of the paper, which is available in the supplementary material.
> We note that for this PO-MEMDP (which is, in turn, a ME-POMDP), the optimal strategy is randomized.
> We also note that when we do not specify an initial belief for the POMDP, we obtain an AB-POMDP.
> We explore that approach to solving ME-POMDPs in Table 3.
>
>
> > Q2. *What does the comparison in Table 3 between ME-POMDPs and AB-POMDPs, both solved with AB-HSVI, entail?*
>
> The results in Table 3 are to compare the naive implementation as suggested in your other question.
> When we transform the ME-POMDP into an explicit POMDP with an unknown initial belief, we can run our AB-HSVI algorithm with a single environment, but it will always have to take the entire state space into account.
> When, instead, we keep the multiple environments, parts of the state space can be discarded once that environment has a probability of zero, leading to a more efficient computation.
> We will clarify the rationale behind the comparison in Table 3 in the paper.
>
>
> > Q3. *In a POSG, why does the fully observing player's policy take the history of states, both players' actions, and observations as the input?*
>
> Regarding the inputs to the fully-observing players' inputs, we note that this assumption is standard for one-sided POSGS [1].
> However, we note that, in our reductions, the fully-observing player only acts at the initial dummy state. Thus, their strategy does not rely on their observing actions, observations, or states.
>
>
>
> > Q4. *Wouldn't we still want to know if the algorithm performs well on some distribution of models rather than on a single randomly generated model?*
>
> *Note: This is the same answer as we give to Reviewer 1 (aQSA) Q3, which we repeat here for readability.*
>
> **In summary:**
> No benchmarks exist for ME-POMDPs, and a straightforward adaptation of existing POMDP benchmarks is not trivial.
> Nevertheless, we provide additional experiments on variations of the RockSample problem.
>
> First, we would like to note that there are no existing benchmarks for ME-POMDPs.
> Removing the initial belief from a POMDP benchmark model often does not result in an interesting problem, as the worst-case environment can be trivial.
> For example, in the original RockSample problem, the initial distribution defines how many rocks are good and bad.
> The environment where all rocks are bad is trivially the worst-case environment.
>
> To create challenging, non-trivial problems for evaluating AB-HSVI, we employed randomization.
> We only use this approach to substitute for the hand-design of models, and thus our experiments are comparable to standard POMDP experiments, where one also evaluates a single instance of each problem (see, for instance, [2], where RockSample was introduced and also used in multiple separate configurations).
>
> Creating challenging problem instances is difficult.
> For example, for the Bird problems with $3$ states, $3$ actions, and $3$ experts, generating models with random seeds $0$ to $99$, we only got $35$ out of $100$ non-trivial PO-MEMDPs, $34$ non-trivial ME-POMDPs, and only $16$ non-trivial MO-POMDPS.
> For this classification, we considered a problem instance trivial if it could be solved within $30$ seconds.
>
> Even though we can generate multiple non-trivial variants of a problem instance, the values and running times are highly dependent on the environments, and these variants are therefore better considered as distinct problems rather than variants that we should compare.
> To demonstrate this, we ran some additional experiments on RockSample instances where the only change was the position of the rocks.
> The number of states, actions, and environments is fixed across the instances.
> The running time between two such variants of a problem instance varied by factors between $3$ and $10$, and we expect this gap to further increase with a larger state space.
>
> We will clarify the use of randomization in our paper and include these experiments to further highlight the effects of the specific model on runtime.
>
>
> #### References
>
> [1] Karel Horák, Branislav Bosanský, Vojtech Kovarík, Christopher Kiekintveld: Solving zero-sum one-sided partially observable stochastic games. Artif. Intell. 316: 103838 (2023)
>
> [2] Trey Smith, Reid G. Simmons: Heuristic Search Value Iteration for POMDPs. UAI 2004

---

> > ### Comment · Reviewer_g1P2 · 2025-08-01
> >
> > All of my clarification questions have been addressed.
> >
> > In response to the weaknesses I mentioned in my original review (presentation and discussion of limitations), the authors promise to "incorporate all other suggestions and directions for future work, paying particular attention to accessibility and readability" and that they "will also highlight the limitations". What this will look like is hard to judge. I would have expected a more detailed description of how they plan to improve the accessibility and readability and, in particular, what the authors take to be the limitations of their method.

---

> > > ### Author Response · Authors · 2025-08-04
> > >
> > > ### Summary
> > >
> > > We acknowledge that more detail is needed regarding how we will improve the clarity of the paper, and below, we provide further details on how we will achieve this improvement. In particular, we will discuss how we aim to achieve the following.
> > >
> > > 1. Add figures to clarify the relationships between the models.
> > >
> > > 2. Incorporate figures to make the experimental results clearer.
> > >
> > > 3. Add natural-language discussions of theoretical results.
> > >
> > > 4. Address the sparsity of the discussion of experimental results.
> > >
> > > 5. Discuss limitations in the paper.
> > >
> > >
> > > ##### 1.
> > >
> > > We acknowledge that the work introduces many models, and to address this, we will add a figure to the introduction of the paper that gives a high-level overview of these models and their relations, as suggested in the original review.
> > > Through this figure, we aim to communicate which models reduce to one another and where readers can find these reductions.
> > > We give a sketch of how part of this figure might look in the equation below.
> > >  $$\text{POMDP} \xrightarrow[]{\text{reduces to}} \text{ME-POMDP} \xleftarrow[\text{Section 4.2}]{\text{interreduce}}\xrightarrow[]{} \text{AB-POMDP} \xrightarrow[\text{Section 4.1}]{\text{reduces to}} \text{POSG}$$
> > > We will also add information to this figure on PO-MEMDPs and MO-POMDPs, and their relationships to ME-POMDPs, as well as references to definitions of all models.
> > >
> > > ##### 2.
> > >
> > > To make the experimental results easier to interpret, we will represent several of our results in plots instead.
> > > The full tables will remain available in the appendix.
> > > More specifically, we will change the following.
> > > - We will replace Table 3 with a bar plot indicating the difference in convergence time and value between the AB-POMDP and ME-POMDP models for each RockSample instance.
> > > - We will add a plot to represent (part of) Table 4, where we will plot the time increase factor out against the potential loss of the naive non-robust policy.
> > > - We will add a plot showing the effect of different environments on the convergence time.
> > > In the rebuttal (Q4), we mentioned new experimental results for comparing the convergence time of multiple RockSample instances.
> > > In these instances, each ME-POMDP shares the same number of states, actions, and environments, but the ME-POMDPs differ in their sets of environments.
> > > Namely, these sets of environments differ in whether the rocks are close to or far from the agent's starting position.
> > > This plot will compare the convergence time of multiple ME-POMDP instances with the same state, action, and environment sizes, where each ME-POMDP instance differs in the set of environments.
> > > - We will also add a new plot showing how run-time scales with state-space size.
> > > As we constructed the new RockSample instances for the previous plot for multiple grid sizes, we can use these same experiments to compare multiple RockSample instances with varying grid sizes, but maintaining the fixed relative positions of the rocks.
> > > We will use these experiments to plot convergence time against the state-space size.
> > > This plot will include multiple curves for the different fixed relative positions of the rocks and the number of environments.
> > > ##### 3.
> > >
> > > In the current version of the paper, we attempted to use the paragraphs before each Theorem statement to describe the upcoming result. Still, we agree that this connection is not clear enough.
> > >
> > > We will rewrite the theoretical sections to more directly communicate what each theorem says.
> > > For example, lines 166-172 currently read like so.
> > >
> > > > *When the set of beliefs is the set $\Delta$(Q) on some subset of states Q, any AB-POMDP is equivalent to
> > > a zero-sum one-sided POSG. In this POSG, the partially observing player is the agent, and they have
> > > the same actions and observations as in the original AB-POMDP. We replace the set of beliefs with a
> > > second player whose action set is the set of states Q. We shall refer to the partially observing player
> > > as the agent, and the fully observing player as nature. By choosing an appropriate distribution over
> > > states, nature can choose a distribution in $\Delta$(Q) against which the agent’s policy is evaluated. The
> > > optimal policy for the agent in this POSG gives an optimal policy in the original AB-POMDP.*

---

> > > > ### Author Response · Authors · 2025-08-04
> > > >
> > > > *(continued)*
> > > >
> > > > To incorporate an explanation of Theorem 1, we will rewrite this paragraph as
> > > > follows (additions and changes in bold-italics).
> > > >
> > > > > *When the set of beliefs is the set $\Delta$(Q) on some subset of states Q, any AB-POMDP is equivalent to
> > > > a zero-sum one-sided POSG,*
> > > > ***and we codify this result in Theorem 1.
> > > > In particular, for an AB-POMDP, Theorem 1 gives a recipe to construct a POSG that allows us to find (optimal) policies for the AB-POMDP.***
> > > > *In this POSG, the partially observing player is the agent, and they have
> > > > the same actions and observations as in the original AB-POMDP. We replace the set of beliefs with a
> > > > second player whose action set is the set of states Q. We shall refer to the partially observing player
> > > > as the agent, and the fully observing player as nature. By choosing an appropriate distribution over
> > > > states, nature can choose a distribution in $\Delta$(Q) against which the agent’s policy is evaluated. The
> > > > optimal policy for the agent in this POSG gives an optimal policy in the original AB-POMDP.*
> > > >
> > > > In particular, in the second sentence of this new paragraph, we give a short natural-language description of Theorem 1's content.
> > > > Throughout the paper, we will add a brief single-sentence natural-language description of each theorem in the preceding paragraph.
> > > >
> > > > ##### 4.
> > > >
> > > > We acknowledge that the current discussion of experimental results is sparse, and we especially thank the reviewer for pointing out that we do not appropriately discuss how the results vary between ME-POMDP, MO-POMDP, and PO-MEMDP models.
> > > > Below, we describe the changes we will make to the discussion of results.
> > > >
> > > > **Convergence time and value across model types.**
> > > > We will add the following discussion of Table 1, which describes the differences in results between ME-POMDP, MO-POMDP, and PO-MEMDP models, to the paper.
> > > >
> > > >
> > > > > For a given ME-POMDP, Table 1 shows how convergence time and value vary when we either (1) fix the observation function to get a PO-MEMDP, or (2) fix the transition function to get a MO-POMDP.
> > > > We observe that, in general, AB-HSVI converges more quickly for MO-POMDPs and returns a higher value, thus indicating that uncertain observation functions are easier to handle than uncertain transitions.
> > > >
> > > > **Relationship between environment generation and difficulty.**
> > > > In the rebuttal, we mentioned new experiments on the rate at which we get challenging models for the Bird problem from randomly generated environments.
> > > > Additionally, we mentioned the new experiments on RockSample instances where the ME-POMDPs share the same number of states, actions, and environments, but each ME-POMDP differs in the set of environments, as we discussed earlier in point 2 of this response.
> > > > We will use these new experiments to discuss the impact of random generation on the ME-POMDP's difficulty.
> > > >
> > > >
> > > > In particular, we will add the following paragraphs to Section 6 of the paper.
> > > >
> > > > > The randomly generated environments have a significant effect on the difficulty of the problems.
> > > > Indeed, for each of the PO-MEMDP, MO-POMDP, and ME-POMDP versions of the Bird problem, we generate 100 random environments.
> > > > We observe only 35 out of 100 non-trivial PO-MEMDPs, 34 non-trivial ME-POMDPs, and only 16 non-trivial MO-POMDPS.
> > > > For this classification, we considered a problem instance trivial if it could be solved within 30 seconds.
> > > >
> > > > > Furthermore, we observe a difference in convergence time of a factor between 3 and 10 for RockSample instances where each ME-POMDP has the same number of states, actions, and environments, but the ME-POMDPs differ in their environment sets.
> > > > Namely, the ME-POMDP instances differ in whether the rocks are close by or far away from the agent's starting position.
> > > > This demonstrates the great effect that the environments have on the difficulty of solving a ME-POMDP.
> > > >
> > > > These paragraphs provide details to support the assertion that "the randomly generated environments have a great effect on the difficulty of the problems".
> > > >
> > > >
> > > >
> > > > ##### 5.
> > > >
> > > > Finally, we present the following draft of a short limitation section appended to the end of the paper.
> > > >
> > > > >    The main limitation of this work is the scalability of AB-HSVI, in particular, the substantial increase in convergence time with the number of environments.  We believe that exploring policy-gradient methods or online-planning methods for ME-POMDPs is a critical next step to ensuring their applicability, and we believe that our theoretical results provide a foundation for this work.
> > > >
> > > >
> > > > We appreciate the reviewer's in-depth feedback on improving the work's readability and would be happy to receive any further suggestions for clarification.

---

> > > > > ### Comment · Reviewer_g1P2 · 2025-08-04
> > > > >
> > > > > Thank you for the detailed response and plans to improve the readability. I have increased my score by one point and think this paper will be a valuable addition to the literature.

---

### Official Review · Reviewer_aQSA · 2025-07-16

**Clarity:** 3
**Significance:** 3
**Originality:** 3
**Rating:** 4
**Confidence:** 3

**Summary:**

This paper addresses the problem of robust planning in POMDPs under discrete model uncertainty, i.e., Multi-Environment POMDPs (ME-POMDPs), where the goal is to compute a single policy that maximizes the worst-case expected reward across a finite set of possible models. They make a theoretical contribution by introducing Adversarial-Belief POMDPs (AB-POMDPs), proving that ME-POMDPs can be reduced to this instance. Algorithmically, they propose the Adversarial-Belief Heuristic Search Value Iteration (AB-HSVI), which integrates linear programming into the popular HSVI point-based solver to adversarially select the belief for value function updates. The effectiveness and scalability of this approach are demonstrated through experiments on POMDP benchmarks.

**Questions:**

1. The scalability of AB-HSVI with respect to the number of environments n and the state-space size |S| appears to be the main bottleneck. While the current work provides a crucial theoretical and algorithmic foundation, have you considered any techniques to mitigate this for larger problems? For instance:
- Could one potentially prune redundant or clearly dominated environments from the set before solving to reduce n?
- Could the core idea of adversarial belief selection be integrated with more scalable online planning methods, such as POMCP, where a complete value function is not maintained?
2. The paper focuses on a classic, point-based solver. The theoretical formulation of ME-POMDPs and AB-POMDPs, however, seems general and valuable. Do you see a path to applying these core ideas in settings where PWLC value functions are intractable? For example, could this be formulated as a two-player zero-sum game solvable with deep reinforcement learning methods, where one agent (the policy) plays against another (an adversary selecting the environment/belief)?
3. The experimental results are reported on a single, randomly generated instance for each problem configuration. This makes it difficult to assess the general performance and variance of the algorithm. Could you comment on how representative you believe these single runs are? For the final version of the paper, I would encourage reporting results averaged over multiple random seeds (e.g., 5-10 runs per configuration), including standard deviations, to provide a more robust empirical validation.
4. The paper provides a proof connecting AB-POMDPs to one-sided POSGs. Does this theoretical connection provide more than just a formal link? For instance, could specific insights or algorithms from the rich POSG literature be leveraged to develop alternative or more efficient solution methods for AB-POMDPs in the future?

**Ethical Concerns:**

["NO or VERY MINOR ethics concerns only"]

**Final Justification:**

I have read other reviews and rebuttals, I think most of my concerns are addressed.

**Quality:**

3

**Strengths And Weaknesses:**

Strengths
1. The paper addresses a well-defined and practically important problem. The assumption that exact model parameters for a POMDP are known is often a bottleneck for real-world applications. The ME-POMDP formulation is an intuitive and relevant model for handling discrete model uncertainty, and the paper's focus on robust, worst-case optimization is appropriate for high-stakes decision-making.
2. The paper presents theoretical analysis. The introduction of AB-POMDPs as a unifying framework looks sound. The theorems that formally establish reductions and equivalences between ME-POMDPs, AB-POMDPs, and one-sided POSGs (Theorems 1-4) can be a good contribution.
3. The proposed AB-HSVI is a technically sound approximate solution method, which builds upon the foundation of HSVI, a popular and effective POMDP algorithm. The integration of a linear program to find the worst-case belief is a direct and elegant way to handle the adversarial nature of the problem within the value iteration framework.


Weaknesses
1. While the experiments analyze scalability, they also confirm that it is a primary limitation. The computational time increases significantly with the number of environments or the problem size. For problems with large state/action space or a large number of potential environments, the current approach may be intractable.
2. The proposed method is rooted in traditional POMDP solvers that rely on maintaining an explicit PWLC value function (a set of alpha-vectors). This representation can become a bottleneck in POMDPs with very large state or observation spaces, which is where modern deep reinforcement learning methods are often applied. The paper does not discuss how these ideas might be extended to work with nonlinear function approximators (e.g., neural networks), or online planing (e.g., POMCP), which limits its applicability to problems outside the scope of classic point-based solvers.
3. They correctly note in the checklist that their algorithm is deterministic. However, the problem instances themselves are generated via randomization. The empirical results would be stronger if, for each configuration (e.g., RS_3,1,3), the authors had generated a small number of random instances (e.g., 5-10) and reported the mean and standard deviation of the results. This would provide a better sense of the method's performance consistency across different models of the same size, rather than just on a single random instance.

---

> ### Author Rebuttal · Authors · 2025-07-31
>
> We would like to thank the reviewer for their helpful comments, suggestions, and questions.
> Based on those, we plan to make the following changes to our paper:
> - Add details on optimization techniques for AB-HSVI.
> - Clarify the use of randomization to find non-trivial benchmarks and include the additional experimental results discussed below to further highlight the effects of specific models on runtime.
> - Incorporate all other suggestions and directions for future work.
>
> We now answer the questions.
>
> > Q1. *Have you considered any techniques to mitigate the bottleneck of the number of environments and the state-space size in AB-HSVI for larger problems?*
>
> We implement two optimization techniques already: sparse representations for vectors and matrices, and pruning of $\alpha$-vectors.
> We use sparse matrices to represent the beliefs and $\alpha$-vectors.
> This optimization is most effective when, during execution, we can infer that a certain environment is impossible.
> For example, a transition may be observed while not being possible in one of the environments, in which an entire part of the matrix will remain zero for the rest of the exploration.
> A sparse representation avoids explicitly storing that part of the matrix.
> Another optimization we implement in AB-HSVI is the pruning of $\alpha$-vectors that are dominated by a single other $\alpha$-vector.
> Using additional pruning methods could improve the efficiency of the algorithm.
> Additional directions to improve AB-HSVI's scalability include keeping track of and reusing previously explored beliefs, and the use of compact representation techniques for the state space, such as Value-directed Compression (VDC) [1].
>
> We will add details on the optimization techniques already used, as well as the additional directions for further optimization, to the paper.
>
> > Q1a. *Could one prune redundant or clearly dominated environments from the set before solving?*
>
> Pruning environments before solving can be done when an environment is a submodel of another environment.
> Examples of situations where such pruning can be performed are:
> - When the transition and observation functions are the same, but one of the reward functions has larger rewards at every state-action pair.
> - When one environment has a sink state, while another environment can continue to accumulate reward.
>
> We did not consider such optimizations in the current work, and thank the reviewer for the interesting suggestion for future improvement.
>
>
> > Q1b. *Could the core idea of adversarial belief be integrated with more scalable online planning methods, such as POMCP?*
>
> A key requirement in online planning methods such as POMCP is that there is a model to simulate.
> In multi-environment POMDPs, it is unclear which environment to simulate, and we need to either choose a POMDP to sample from or assume a distribution over these POMDPs.
> If we were to select an initial environment distribution to sample from and update the distribution along the way, we are changing the environment we are sampling from, essentially breaking the assumption that the environment is fixed once nature chooses it.
>
> One promising direction, however, comes from the recent work on online planning methods for POSGs [2], which, in combination with our ME-POMDP to POSG reduction, suggests that we could explore the use of adversarial beliefs in POMCP.
>
>
> > Q2. *Do you see a path to applying the core ideas of ME-POMDPs and AB-POMDPs in settings where PWLC value functions are intractable?*
>
>
> In this work, we solve multi-environment POMDPs by constructing piecewise linear convex value functions, and we attempt to lessen the computational burden by providing point-based approximation methods.
>
> However, we acknowledge that even these methods are intractable for large mere POMDPs already, and that there are indeed interesting directions for the future.
> One possible approach to lessen the computational burden is to use policy-gradient methods in a game setting [3,4].
> Such methods could prove interesting both from a computational efficiency perspective and a theoretical perspective.
>
> > Q3. *Could you comment on how representative you believe the single runs on the randomly generated instances for each problem configuration are?*
>
>
> **In summary:**
> No benchmarks exist for ME-POMDPs, and a straightforward adaptation of existing POMDP benchmarks is not trivial.
> Nevertheless, we provide additional experiments on variations of the RockSample problem.
>
> First, we would like to note that there are no existing benchmarks for ME-POMDPs.
> Removing the initial belief from a POMDP benchmark model often does not result in an interesting problem, as the worst-case environment can be trivial.
> For example, in the original RockSample problem, the initial distribution defines how many rocks are good and bad.
> The environment where all rocks are bad is trivially the worst-case environment.
>
> To create challenging, non-trivial problems for evaluating AB-HSVI, we employed randomization.
> We only use this approach to substitute for the hand-design of models, and thus our experiments are comparable to standard POMDP experiments, where one also evaluates a single instance of each problem (see, for instance, [5], where RockSample was introduced and also used in multiple separate configurations).
>
> Creating challenging problem instances is difficult.
> For example, for the Bird problems with $3$ states, $3$ actions, and $3$ experts, generating models with random seeds $0$ to $99$, we only got $35$ out of $100$ non-trivial PO-MEMDPs, $34$ non-trivial ME-POMDPs, and only $16$ non-trivial MO-POMDPS.
> For this classification, we considered a problem instance trivial if it could be solved within $30$ seconds.
>
> Even though we can generate multiple non-trivial variants of a problem instance, the values and running times are highly dependent on the environments, and these variants are therefore better considered as distinct problems rather than variants that we should compare.
> To demonstrate this, we ran some additional experiments on RockSample instances where the only change was the position of the rocks.
> The number of states, actions, and environments is fixed across the instances.
> The running time between two such variants of a problem instance varied by factors between $3$ and $10$, and we expect this gap to further increase with a larger state space.
>
> We will clarify the use of randomization in our paper and include these experiments to further highlight the effects of the specific model on runtime.
>
>
> > Q4. *Could specific insights or algorithms from the rich POSG literature be leveraged to develop alternative or more efficient solution methods for AB-POMDPs in the future?*
>
> AB-HSVI uses classical $\alpha$-vector techniques,  and could be considered a specialized ME-POMDP analogue to the HSVI-type algorithms used in POSG papers such as [6].
> We expect that other POSG methods can be adapted to the multi-environment setting, and we believe these adaptations to be an interesting avenue for future work.
> In particular, various works study the problem of reinforcement learning for partially-observable Markov games [7,8,9], which would be interesting to explore in the special case of ME-POMDPs.
> Similarly, as discussed for Q1b, recent work establishes online planning algorithms for POSGs, and it would be interesting to explore simplifications of these methods for ME-POMDPs.
>
> #### References
>
> [1] Pascal Poupart, Craig Boutilier: Value-Directed Compression of POMDPs. NIPS 2002
>
> [2] Tyler J. Becker, Zachary Sunberg: Bridging the Gap between Partially Observable Stochastic Games and Sparse POMDP Methods. AAMAS 2025
>
> [3] Maris F. L. Galesloot, Roman Andriushchenko, Milan Ceska, Sebastian Junges, Nils Jansen: rfPG: Robust Finite-Memory Policy Gradients for Hidden-Model POMDPs. arXiv preprint arXiv:2505.09518 (2025)
>
> [4] Constantinos Daskalakis, Dylan J. Foster, Noah Golowich: Independent Policy Gradient Methods for Competitive Reinforcement Learning. NeurIPS 2020
>
> [5] Trey Smith, Reid G. Simmons: Heuristic Search Value Iteration for POMDPs. UAI 2004
>
> [6] Karel Horák, Branislav Bosanský, Vojtech Kovarík, Christopher Kiekintveld: Solving zero-sum one-sided partially observable stochastic games. Artif. Intell. 316: 103838 (2023)
>
> [7] Qinghua Liu, Csaba Szepesvári, Chi Jin: Sample-Efficient Reinforcement Learning of Partially Observable Markov Games. NeurIPS 2022
>
> [8] Awni Altabaa, Zhuoran Yang: On the Role of Information Structure in Reinforcement Learning for Partially-Observable Sequential Teams and Games. NeurIPS 2024
>
> [9] Xiangyu Liu, Kaiqing Zhang: Partially Observable Multi-agent RL with (Quasi-)Efficiency: The Blessing of Information Sharing. ICML 2023

---

> > ### Author Response · Authors · 2025-08-06
> >
> > We thank the reviewer again for their detailed comments and questions, and hope our response is satisfactory. We would be happy to elaborate if needed.

---

> > ### Comment · Reviewer_aQSA · 2025-08-08
> > **Thanks for your response**
> >
> > I have read other reviews and rebuttals, I think most of my concerns are addressed. Thank you for the detailed response.

---

### Decision · Program_Chairs · 2025-09-17

**Decision:**

Accept (poster)

**Comment:**

All reviews of this work leaned towards acceptance, with two recommendations to accept [g1P2,xCNX] and two recommendations for borderline acceptance [aQSA,sP8H].

Reviewers appreciated the work along several aspects:

+ The problem was considered well-defined and practically important and a known bottleneck for real-world applications [aQSA]
+ The approach was considered to have a convincing argument for novelty [xCNX]
+ The method formulation was considered intuitive and relevant for handling discrete model uncertainty [aQSA]
+ The solution method was considered clear [xCNX] and technically sound [aQSA,xCNX] and well motivated [sP8H] and the theoretical depth was appreciated [sP8H]
+ The focus on robust worst-case optimisation was appreciated [aQSA]
+ The theoretical analysis was considered sound [aQSA]
+ The empirical evaluation metholdology was considered clear and reproducible [xCNX]

A number of concerns were raised in the reviews, including computational complexity and impact of the randomly generated problems, which the authors responded to in the rebuttals:

- Computation time and intractability for large state/action spaces or numerous potential environments was a concern [aQSA]; similarly, theoretical analysis of time or space complexity was desired [sP8H], and discussion of computation cost of changing individual factors [xCNX]; authors did not provide a full analysis but provided some discussion and commented on runtime in a RockSample environment
- Proof or empirical evaluations showing the algorithm converges was desired [xCNX]; authors did not provide a proof but gave discussion of a path towards showing it.
- Feasibility of solving for the robust policy was considered unclear [xCNX]
- A more substantial ablation study was desired [xCNX]
- Maintaining an explicit PWLC value function representation was considered a bottleneck, and discussion of extending to nonlinear function approximations was desired [aQSA]
- Reporting of means and standard deviations across randomly generated problems was desired [aQSA,Sp8H], and similarly, performance on a distribution of models was desired [g1P2], and discussion of how randomness in benchmark generation may affect results was desired [sP8H]; authors argued the randomisation is nontrivial and results should be treated as distinct problems.
- Mitigation of the scalability bottleneck by pruning or online planning methods was raised as a possibility [aQSA]; authors provided some discussion on optimisation techniques already used, considered some pruning approaches future work, and discussed obstacles and possibilities of online planning.
- Applying the core ideas in settings where PWLC value functions are intractable was raised [aQSA]; authors commented on this as a direction of future work.
- Performance versus robust POMDPs involving continuous or convex uncertainty sets was questioned [sP8H]; authors commented available robust POMDP solvers would need a rectangularity assumption not available in the authors' problems making such solvers ineffective.
- A question of further impact of the theoretical connections between AB-POMDPs and one-sides POSGs was raised [aQSA]; authors considered this future work.
- Questions were raised on whether one could use a naive approach solving a POMDP with additional unobserved variables directly; and on meaning of the comparison in Table 3. Authors clarified the naive approach would not yield robust policies, and clarified the rationale of Table 3.
- Clarifying assumptions of same units between models in the reward function was raised [xCNX]; authors planned to discuss it.
- Clarifying the impact of differences between models (e.g. if they disagree on the objective) was desired [xCNX]; authors briefly discussed this but seemed to leave it for future work.
- The message of Table 4 was considered unclear due to the roles of the robust versus optimal policy [xCNX]; authors planned to clarify it.
- Several concerns on readability were raised [g1P2] including clarity of the parts after the introduction, lack of figures, lack of intuitive descriptions of theorems in sections 4 and 5, lack of explanation of some symbols, and sparsity of discussion of the results. Authors promised to clarify the relevant parts of the paper.


After the rebuttals, [aQSA] considered most of their concerns addressed. Reviewer [g1P2] also seemed to feel the concerns were mostly addressed but noted that the proposed changes were hard to judge without detail; authors the provided some plans for the detail. Reviewer [xCNX] considered all the questions and concerns addressed. Reviewer [sP8H] seemed to think concerns were addressed.

Overall, it seems although there are remaining concerns of e.g. computational complexity, there is enough appreciation of the current content that the work may be suitable for presentation at NeurIPS.